# Learning to Compose: Improving Object Centric Learning by Injecting Compositionality

**Whie Jung, Jaehoon Yoo, Sungjin Ahn, Seunghoon Hong**
School of Computing, KAIST
`{whieya, wogns98, sungjin.ahn, seunghoon.hong}@kaist.ac.kr`

## Abstract

Learning compositional representation is a key aspect of object-centric learning as it enables flexible systematic generalization and supports complex visual reasoning. However, most of the existing approaches rely on auto-encoding objective, while the compositionality is implicitly imposed by the architectural or algorithmic bias in the encoder. This misalignment between auto-encoding objective and learning compositionality often results in failure of capturing meaningful object representations. In this study, we propose a novel objective that explicitly encourages compositionality of the representations. Built upon the existing object-centric learning framework (*e.g.*, slot attention), our method incorporates additional constraints that an arbitrary mixture of object representations from two images should be valid by maximizing the likelihood of the composite data. We demonstrate that incorporating our objective to the existing framework consistently improves the objective-centric learning and enhances the robustness to the architectural choices.

## 1 Introduction

As the world is highly compositional in nature, relatively few composable units, such as objects or words, can describe infinitely many observations. Consequently, human intelligence has evolved to recognize the environment as a combination of composable units, (*e.g.,* objects) which enables rapid adaptation to unseen situations by recomposing the already learned concepts (Spelke, 1990; Lake et al., 2017). Mimicking human intelligence, perceiving environment with composable abstractions have shown consistent improvement in tasks related to systematic generalization (Kuo et al., 2021; Bogin et al., 2021; Rahaman et al., 2021), and visual reasoning tasks (D'Amario et al., 2021; Assouel et al., 2022) compared to distributed counterparts.

Inheriting this spirit, object-centric learning (Burgess et al., 2019; Greff et al., 2019; Engelcke et al., 2020; Locatello et al., 2020) aims to discover a composable abstraction purely from data without external supervision. Instead of depicting a scene with a distributed representation, it decomposes the scene into a set of latent representations, where each latent is expected to capture a distinct object. To discover such representation in an unsupervised manner, most existing works employed an auto-encoding framework, where the model is trained to encode the scene into a set of representations and decode them back to the original image.

However, the auto-encoding objective is inherently insufficient to learn compositional representation, since maximizing the reconstruction quality does not necessarily requires the object-level disentanglement. To reduce this gap, the existing works incorporate strong inductive biases to further regularize the encoder, such as architectural bias (Locatello et al., 2020) or algorithmic bias (Burgess et al., 2019; Lin et al., 2020; Jiang et al., 2020). However, it has been widely observed that these methods are highly sensitive to the choice of hyper-parameters, such as encoder and decoder architectures, and a number of slots, often resulting in suboptimal decompositions by position or partial attributes (Singh et al., 2022a; Sajjadi et al., 2022; Jiang et al., 2023) instead of objects. Finding the optimal model configuration is also not straightforward in practice due to the missing object labels.

In this work, we present a novel objective that directly optimizes the compositionality of representations. Based upon the auto-encoding framework, our method extracts object representations independently from two distinct images and simulates their composition by the random mixture.

The composite representation is rendered to an image by the decoder, whose likelihood is evaluated by the generative prior. The encoder is then jointly optimized to minimize the reconstruction error of the individual images to encode relevant information of the scene (*auto-encoding path*) while maximizing the likelihood of the composite image to ensure the compositionality of the representation (*composition path*). Overall, our method can be viewed as extending the conventional auto-encoding approach with an additional regularization on compositionality. We show that directly injecting compositionality this way significantly boosts the overall quality of object-centric representations and robustness in training.

Our contributions are as follows. **(1)** We introduce a novel objective that explicitly encourages compositionality of representations. To this end, we investigate strategies to simulate the compositional construction of an image and propose a learning objective for maximizing the likelihood of the composite images. **(2)** We evaluate our framework on four datasets and verify that our model consistently surpasses auto-encoding based baselines by a substantial margin. **(3)** We show that our objective enhances the robustness of object-centric learning on three major factors, such as number of latents, encoder and decoder architectures.

## 2 PRELIMINARY

**Problem setup** Object-centric learning aims to discover a set of composable representations from an unlabeled image. Formally, given an image $\mathbf{x} \in \mathbb{R}^{H \times W \times C}$ represented by either RGB pixels or feature from the pre-trained encoder, the objective is to extract the set $\mathbf{S} = \{\mathbf{s}_1, \ldots, \mathbf{s}_N\} = E_\theta(\mathbf{x})$, where each element $\mathbf{s}_i \in \mathbb{R}^D$ corresponds to the representation of a composable concept (*e.g.*, an object). Since object concepts should emerge from the data without supervision, a typical approach is to use an auto-encoding framework to formulate the learning process. Formally, the object-centric encoder $E_\theta : \mathbb{R}^{H \times W \times C} \to \mathbb{R}^{N \times D}$ is trained jointly with a decoder $D_\phi : \mathbb{R}^{N \times D} \to \mathbb{R}^{H \times W \times C}$ by minimizing the reconstruction loss.

$$\mathcal{L}_{\text{AE}}(\theta, \phi) = \mathbb{E}_{\mathbf{x}} \left[ d(\mathbf{x}, D_\phi(E_\theta(\mathbf{x}))) \right], \tag{1}$$

where $d$ is a distance metric (*e.g.,* MSE).

**Slot Attention Encoder** $E_\theta$ Since the auto-encoding objective is insufficient to learn highly structured representation, the existing approaches incorporate a strong architectural bias in the encoder $E_\theta$ to guide the object-level disentanglement in $\mathbf{S}$. Among many variants, we consider Slot Attention encoder Locatello et al. (2020) due to its popularity and generality. It employs a dot-product attention mechanism between a query (slot) and a key (input), where normalization is applied over the slots by:

$$\mathbf{A}(\mathbf{x}, \mathbf{S}) = \underset{N}{\text{softmax}} \left( \frac{k(\mathbf{z}) \cdot q(\mathbf{S})^T}{\sqrt{D}} \right) \in \mathbb{R}^{M \times N}, \tag{2}$$

where $\mathbf{z} = f_\theta(\mathbf{x}) \in \mathbb{R}^{M \times D'}$ is a flattened input feature encoded by CNN encoder $f_\theta$, and $k, q$ represents linear projection matrices. Note that softmax operation is normalized in the query (slots) direction, inducing competition among slots. Based on Equation 2, the slots are iteratively refined by:

$$\mathbf{S}^{(n+1)} = \text{GRU}(\mathbf{S}^{(n)}, \text{Normalize}(\mathbf{A}(\mathbf{x}, \mathbf{S}^{(n)})^T \cdot v(\mathbf{z}))), \quad \mathbf{S}^{(0)} \sim \mathcal{N}(\mu, \text{diag}(\sigma)). \tag{3}$$

Here, $\mathbf{S}^{(n)}$ denotes the slot representation after $n$ iterations, $\mu, \sigma$ are learnable parameters characterizing the distribution of the initial slots, $v$ is a linear projection matrix, and Normalize$(\cdot)$ is a weighted mean operation introduced by Locatello et al. (2020) to improve stability of the attention.

**Slot Decoder** $D_\phi$ While the architectural choice for $D_\phi$ is not constrained to a specific form in principle, subsequent works (Singh et al., 2022a; Jiang et al., 2023) have empirically found that the choice of the decoder crucially impacts the quality of the object-centric representation. Locatello et al. (2020) proposed a pixel-mixture decoder that renders each slot independently into pixels and combines them with alpha-blending. Although slot-wise decoding provides a strong incentive for the encoder to capture distinct objects in each slot, its limited expressiveness hinders its application to complex scenes. To address this issue, Singh et al. (2022a) employed Transformer decoder that takes the entire slots $\mathbf{S}$ as an input and produces an image in an autoregressive manner. By modeling

the complex interactions among the slots, it has shown great improvements in slot representation learning even in complex scenes.

Recently, Jiang et al. (2023) employed a diffusion model for the slot decoder. Instead of directly reconstructing an input image $\mathbf{x}$, it optimizes the auto-encoding of Equation 1 via denoising objective (Ho et al., 2020) by:

$$\mathcal{L}_{\text{Diff}}(\theta, \phi) = \mathbb{E}_{\epsilon \sim \mathcal{N}(\mathbf{0}, \mathbf{I}), t \sim U(0,1)} \left[ w(t) \cdot \| D_\phi(\mathbf{x}_t, t, \mathbf{S} = E_\theta(\mathbf{x})) - \epsilon \|^2 \right], \tag{4}$$

where $\mathbf{x}_t = \sqrt{\bar{\alpha}_t} \mathbf{x} + \sqrt{1 - \bar{\alpha}_t}$ is an corrupted image of an input $\mathbf{x}$ by the forward diffusion process at step $t$, $\bar{\alpha}_t = \prod_i^t (1 - \beta_i)$ is a schedule function, and $w(t)$ is the weighting parameter. In practice, the diffusion decoder is implemented based on UNet architecture (Rombach et al., 2022), where each layer consists of a CNN-layer followed by a slot-conditioned Transformer. Once trained, the decoder generates an image $\mathbf{x} \sim p_\phi(\mathbf{x}|\mathbf{S})$ using iterative denoising, starting from the random Gaussian noise (Ho et al., 2020; Rombach et al., 2022). Employing a diffusion decoder significantly enhances object-centric representation and generation quality compared to previous arts especially in complex scenes Jiang et al. (2023).

## 2.1 LIMITATIONS

While the slot attention with auto-encoding objectives has shown promise in object-centric learning, its success highly depends on the model architectures, such as number of slots and architectures of the encoder and decoder, where suboptimal configuration often leads to dividing the scenes into tessellations (Singh et al., 2022a; Sajjadi et al., 2022) and objects into the parts (Jiang et al., 2023). However, the optimal model configuration varies depending on the datasets, and discovering them through cross-validation is practically infeasible due to the missing object labels in an unsupervised setting. We argue that such instability is primarily because the auto-encoding objective is inherently misaligned with the one for object-centric learning, since the former guides the encoder only to minimize the information loss on the input, while the latter demands the object-level disentanglement in the representation, potentially sacrificing the reconstruction quality. This motivates us to seek an alternative approach that directly encourages object-level disentanglement in the objective function instead of designing architectural biases.

## 3 LEARNING TO COMPOSE

Our goal is to improve object-centric learning by modifying its objective function to be more directly aligned with learning compositional slot representation than the auto-encoding loss. Our main intuition is that arbitrary compositions of object representation are likely to yield another *valid* representation. To realize this intuition, our framework is designed to generate composite images by mixing slot representations from two images and maximize their validity measured by the data prior.

Figure 1 illustrates the overall framework of our method. Our framework is built upon the conventional object-centric learning that learns both the slot encoder and decoder by the auto-encoding path on individual images (Section 2). To impose compositionality on slot representation, we incorporate an additional *composition path* that constructs a composite slot representation from two images by the mixing strategy (Section 3.1) and assesses the quality of the image generated from the mixed slots by the generative prior (Section 3.2). This way, the auto-encoding path ensures that each slot contains the relevant information of an input image, while such slots are constrained to capture composable components of the scenes (*e.g.*, objects) by regularizing the encoder through the composition path.

## 3.1 MIXING STRATEGY FOR COMPOSING SLOT REPRESENTATION

Given $\mathbf{S}^1, \mathbf{S}^2 \in \mathbb{R}^{N \times D}$ extracted from two distinct images $\mathbf{x}^1, \mathbf{x}^2$, we construct their composite slot representation $\mathbf{S}^c \in \mathbb{R}^{N \times D}$ by

$$\mathbf{S}^c = \pi(\mathbf{S}^1, \mathbf{S}^2), \tag{5}$$

where $\pi(\cdot, \cdot)$ denotes a composition function of two sets. The primary role of the composition function is to simulate potential combinations of slot-wise compositions. Since our goal is to maximize the compositionality of unseen slot combinations, the composition function should be capable of exploring a broad range of compositional possibilities. Below, we introduce simple instantiations of such function.

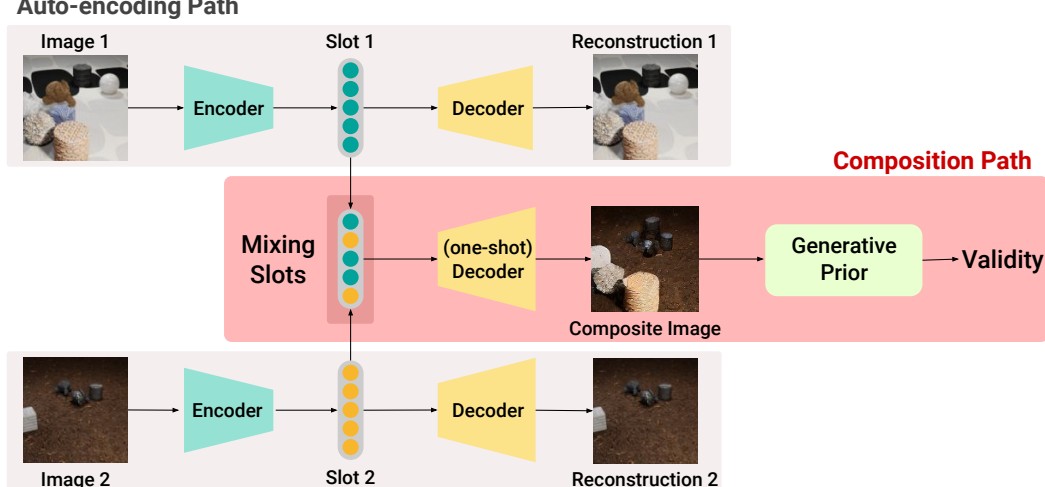

Figure 1: **Overview of our method.** Our framework consists of two paths: an auto-encoding path and a composition path. The auto-encoding path ensures slot representations encode relevant information about an image. In contrast, the composition path encourages the compositionality of the representations by constructing the composite representation through the mixture of slots from two separate images (Section 3.1), and assessing the quality of the composite image by the generative prior (Section 3.2). The encoder is jointly optimized by both paths.

**Random Sampling**    In this approach, we randomly sample $N$ slots among $2N$ slots *i.e.*, $\mathbf{S}^c \overset{N}{\sim} (\mathbf{S}^1 \cup \mathbf{S}^2)$. As it explores over all of the possible combinations, this composition function encourages the slot representation itself to be highly composable to generate valid images for any combinations. On the other hand, it may produce invalid combinations of slots on rare occasions, *e.g.*, omitting the background slots or sampling two objects placed in the same location.

**Sharing Slot initialization**    One way to mitigate such suspicious compositions is to constrain $\mathbf{S}^c$ to be valid composition of the scene. However, strictly ensuring this constraint is non-trivial due to the stochastic nature of slot attention *i.e.*, each slot is sampled stochastically from its underlying distribution and the association between the slots and scenes varies depending on the initialization. Instead, we adopt a rather simple approach that employs the identical slot initialization $\mathbf{S}^{(0)}$ in Equation 3 for two images, and sample the exclusive set of slots. Formally, let $I_1$ and $I_2$ be a random partition of slot indices *i.e.*, $I_1 \cup I_2 = \{1, ..., N\}, I_1 \cap I_2 = \emptyset$. Then we construct the composite slot by $\mathbf{S}^c = \mathbf{S}^1_{I_1} \cup \mathbf{S}^2_{I_2}$, where $\mathbf{S}^1$ and $\mathbf{S}^2$ are slots extracted by Equation 3 from $\mathbf{x}^1$ and $\mathbf{x}^2$, respectively, which are initialized with the same $\mathbf{S}^{(0)}$. The underlying intuition is that the slot initialization is reasonably correlated with the objects it captures (Figure 6), hence sampling from exclusive slots is likely to be valid scenes than the random sampling.

## 3.2    MAXIMIZING LIKELIHOOD OF THE COMPOSITE IMAGE

Given the composite slot $\mathbf{S}^c$ obtained by the previous section, our next step is quantifying its validity *i.e.*, measuring how valid the composition of two image slots is. To this end, we decode it back to an image by $\mathbf{x}^c = D_\phi(\mathbf{S}^s)$ and measure the likelihood of the image using the generative prior $p(\mathbf{x}^c)$.

**Generative Prior**    To model the generative prior $p(\mathbf{x}^c)$, we opt for a diffusion model (Ho et al., 2020) due to its excellence in generation quality and mode coverage (Xiao et al., 2022). The latter is especially important in our framework since the model evaluates the prior over potentially out-of-distribution samples generated by the composition (Section 3.1). Instead of introducing an additional pre-trained diffusion model, we employ the diffusion-based decoder in the auto-encoding path (Section 2), and reuse it as a generative prior. This way, our decoder $D_\phi$ is trained by minimizing the reconstruction loss by denoising objective in Equation 4, while serving as a generative prior in the composition path. It greatly improves the parameter-efficiency and memory, and the need for pre-trained generative prior per dataset.

**Maximizing** $p(\mathbf{x}^c)$  Given the generative prior, we maximize the likelihood $p(\mathbf{x}^c)$ with respect to $\mathbf{x}^c$ in the composition path. Since $\mathcal{L}_{\text{Diff}}$ in Equation 4 is minimizing the upper bound of negative log likelihood of $x^c$ (Ho et al., 2020), minimizing $\mathcal{L}_{\text{Diff}}$ with respect to $\mathbf{x}^c$ leads to the maximization of the likelihood $p(\mathbf{x}^c)$. However, computing the gradient of $\mathcal{L}_{\text{Diff}}$ requires expensive computation of Jacobian maxtrix of the decoder and it often degrades the overall training stability. Following (Poole et al., 2022), the gradient of $\mathcal{L}_{\text{Diff}}$ with respect to $\theta$ can be approximated as:

$$\nabla_\theta \mathcal{L}_{\text{Prior}}(\theta) = \mathbb{E}_{t,\epsilon}[w(t)(D_\phi(\mathbf{x}_t^c, t, \mathbf{S}^c) - \epsilon)\frac{\partial \mathbf{x}^c}{\partial \theta}]. \tag{6}$$

where $\epsilon \sim \mathcal{N}(\mathbf{0}, \mathbf{I})$ is a noise, $t \sim \mathcal{U}(t_{\min}, t_{\max})$ is a timestep, respectively, $w(t)$ is a weighting function dependent to $t$, and $\mathbf{x}_t^c = \sqrt{\bar{\alpha}_t}\mathbf{x}^c + \sigma_t \epsilon$ is a corrupted image of $\mathbf{x}^c$ from forward diffusion process. By updating the encoder parameters $\theta$ with $\nabla_\theta \mathcal{L}_{\text{Prior}}$, $\mathbf{x}^c$ is guided toward high probability density region following the diffusion prior. Note that optimization of the Equation 6 is with only respect to the encoder parameter while fixing the decoder. It prevents suspicious collaboration between the encoder and decoders in generating composite images from suboptimal slots.

**Surrogate One-Shot Decoder**  As discussed earlier, our framework exploits the diffusion model $D_\phi$ as a decoder and generative prior in the auto-encoding and composition paths, respectively. One drawback is that the diffusion decoder requires an iterative denoising process to generate the composite image $\mathbf{x}^c$, which takes significant time and makes the backpropagation through the decoder non-trivial. To address this problem, we employ a one-shot decoder $D_\psi$ as a surrogate for $D_\phi$ to support fast and differentiable decoding of $\mathbf{x}^c$. [1]

We employ a bidirectional Transformer (Devlin et al., 2019) that takes the composite slot $\mathbf{S}^c$ and the learnable mask tokens $\mathbf{m} \in \mathbb{R}^{HW \times C}$ as input, and produces the composite image by a single forward process by $\mathbf{x}^c = D_\psi(\mathbf{m}, \mathbf{S}^c)$. The decoder is trained along with the auto-encoding path by:

$$\mathcal{L}_{\text{Recon}}(\theta, \psi) = ||D_\psi(\mathbf{m}, E_\theta(\mathbf{x})) - \mathbf{x}||^2. \tag{7}$$

Note that the generation quality of the one-shot decoder $D_\psi$ is behind the powerful diffusion decoder $D_\phi$, and serves only to compute the $\mathbf{x}^c$ in Equation 6. We observe that such weak decoder is sufficient to compute the meaningful gradient through the Equation 6, presumably because the gradients are accumulated over various noise levels $t$.

### 3.3  Learning Objective

In this section, we summarize the overall framework and objective function. Our framework consists of two paths; auto-encoding path and composition path. In auto-encoding path, encoder $E_\theta$ and two different decoders $D_\phi, D_\psi$ are trained to minimize auto-encoding objective in Equation 4 and Equation 7. In composition path, we first extract $\mathbf{S}^c$ with Equation 5 and generate $\mathbf{x}^c$ with the deterministic decoder $D_\psi$, and update the encoder to maximize the Equation 6 while fixing decoders $D_\phi$ and $D_\psi$. We find that incorporating an additional regularization term on the slot attention mask is helpful in enhancing object-centric representations:

$$\mathcal{L}_{\text{Reg}}(\theta) = \mathbf{A}^1 \cdot \text{sg}(||\mathbf{x}^1 - \mathbf{x}^c||^2) + \mathbf{A}^2 \cdot \text{sg}(||\mathbf{x}^2 - \mathbf{x}^c||^2), \tag{8}$$

where $\mathbf{A}^1 = \mathbf{A}(\mathbf{x}^2, \mathbf{S}^{1^{(n)}}), \mathbf{A}^2 = \mathbf{A}(\mathbf{x}^1, \mathbf{S}^{2^{(n)}})$ are attention masks from the last iteration of slot attention for $\mathbf{x}^1, \mathbf{x}^2$ (Equation 2), respectively, and $\text{sg}(\cdot)$ denotes stop-gradient operator. It encourages the source and the composite images to be consistent over the object area captured by the slots, enhancing the content-preserving composition. The overall objective is then formulated as follow:

$$\mathcal{L}_{\text{Total}}(\theta, \phi, \psi) = \lambda_{\text{Prior}}\mathcal{L}_{\text{Prior}}(\theta) + \lambda_{\text{Diff}}\mathcal{L}_{\text{Diff}}(\theta, \phi) + \lambda_{\text{Recon}}\mathcal{L}_{\text{Recon}}(\theta, \psi) + \lambda_{\text{Reg}}\mathcal{L}_{\text{Reg}}(\theta) \tag{9}$$

where $\lambda_{\text{Prior}}, \lambda_{\text{Diff}}, \lambda_{\text{Recon}}, \lambda_{\text{Reg}}$ are hyperparameters for controlling the importance of each term. We empirically find that $\lambda_{\text{Prior}} = \lambda_{\text{Diff}} = \lambda_{\text{Recon}} = 1.0, \lambda_{\text{Reg}} = 0.25$ generally works well and use it throughout the experiments.

---

[1] We also consider one-step denoising result of the diffusion decoder using Tweedie's formula (Stein, 1981; Robbins, 1992) but observe severe degradation in performance due to its inferior quality.

## 4 RELATED WORK

**Object-centric learning**    The most dominant paradigm of object-centric learning is employing the auto-encoding objective (Burgess et al., 2019; Greff et al., 2019; Engelcke et al., 2020; 2021; Lin et al., 2020; Jiang et al., 2020; Eslami et al., 2016; Crawford & Pineau, 2019). To guide the model to learn structured representation under reconstruction loss, Locatello et al. (2020) introduces Slot Attention, where each slot is iteratively refined with dot-product attention mechanism normalized in slot direction, inducing competition between the slots. Follow-up studies (Singh et al., 2022a; Seitzer et al., 2022; Sajjadi et al., 2022) demonstrate that Slot Attention with an auto-encoding objective has the potential to attain object-wise disentanglement even in complex scenes. Nonetheless, auto-encoding alone often involves training instability, which leads to attention-leaking problem (Kim et al., 2023), or dividing the scene into Voronoi tessellations (Sajjadi et al., 2022; Jiang et al., 2023). To overcome such challenges, there have been a few attempts on revising the learning objective such as replacing image reconstruction loss with denoising objective (Jiang et al., 2023; Wu et al., 2024) or contrastive loss (Hénaff et al., 2022; Wen et al., 2022). Nevertheless, these approaches still do not impose direct learning of object-centric representations.

**Generative Prior**    There are increasing interests in exploiting the knowledge pre-trained from generative prior to various applications such as solving inverse problems (Chung et al., 2023), guidance in conditional generation (Graikos et al., 2022; Liu et al., 2023), and image manipulations (Ruiz et al., 2023a; Zhang et al., 2023; Ruiz et al., 2023b). One prominent approach in this direction is text-to-3D Generation, where a large-scale pre-trained 2D diffusion model (Rombach et al., 2022; Saharia et al., 2022) is leveraged to generate realistic 3D data without ground-truth (Wang et al., 2023a; Lin et al., 2023; Metzer et al., 2023; Wang et al., 2023b). The seminal work by (Poole et al., 2022) formulates a loss based on a probability density distillation to distill a pre-trained 2D image prior to a 3D model. Back-propagating the loss through a randomly initialized 3D model, *e.g.*, NeRF (Mildenhall et al., 2020), the model gradually updates to generate high-fidelity 3D renderings. Inspired by this line of work, we employ a generative model in our approach to maximize the validity of the given images.

## 5 EXPERIMENT

**Implementation Details**    We base our implementation on existing frameworks (Singh et al., 2022a; Jiang et al., 2023). We employ the features from the pre-trained auto-encoder[2] to represent an image. For the slot encoder, we employ the CNN based on UNet architecture (Singh et al., 2022b; Jiang et al., 2023) to produce a high-resolution attention map. Also, we employ an implicit Slot Attention (Chang et al., 2022) to stabilize the iterative refinement process in slot attention. For the slot mixing strategy, we opt for a sampling with sharing slot initializations for all the experiments unless specified, since it shows slightly better performance than the random sampling strategy. When we compute $\mathcal{L}_{\text{Prior}}$ (Equation 6), we use $t_{\min} = 0.02, t_{\max} = 0.5$ following a recent report in (Wang et al., 2023b) that employing too high noise level impairs the optimization.

**Datasets**    We validate our method on four datasets. CLEVRTex (Karazija et al., 2021) consists of various rigid objects with homogeneous textures. MultiShapeNet (Stelzner et al., 2021) includes more complex and realistic furniture objects. PTR (Hong et al., 2021) and Super-CLEVR (Li et al., 2023) contain objects composed of multi-colored parts and textures. All of the datasets are center-cropped and resized to 128x128 resolution images. .

**Baselines**    We compare our method against two strong baselines in the literature, SLATE (Singh et al., 2022a) and LSD (Jiang et al., 2023), which employ autoregressive Transformer and diffusion-based decoders, respectively. Note that our method without composition path reduces to LSD. For a fair comparison, we employ the same encoder architecture based on slot attention (Locatello et al., 2020) in all compared methods including ours. For LSD and our method, we employ the same pre-trained auto-encoder (Rombach et al., 2022) to represent an input image. Since SLATE runs on discrete features, we employ the features from the pre-trained VQGAN model (Esser et al., 2021) and denote it as SLATE+. All baselines including ours are trained for 200K iterations.

**Evaluation Metrics**    Following the previous works (Jiang et al., 2023; Singh et al., 2022a;b; Chang et al., 2022), we report the unsupervised segmentation performance with three measures: Adjusted

---

[2]https://huggingface.co/stabilityai/sd-vae-ft-ema-original

Table 1: **Comparison results on unsupervised object segmentation.** We evaluate the how well the slot attention masks coincide with the ground-truth objects using FG-ARI, mIoU, and mBO (The higher is better). All results are evaluated on held-out validation set.

(a) CLEVRTex

| Model | FG-ARI | mIoU | mBO |
|---|---|---|---|
| SLATE+ | 71.29 | 52.04 | 52.17 |
| LSD | 76.44 | 72.32 | 72.44 |
| Ours | **93.06** | **74.82** | **75.36** |

(b) MultiShapeNet

| Model | FG-ARI | mIoU | mBO |
|---|---|---|---|
| SLATE+ | 70.44 | 15.55 | 15.64 |
| LSD | 67.72 | 15.39 | 15.46 |
| Ours | **89.8** | **59.21** | **59.4** |

(c) PTR

| Model | FG-ARI | mIoU | mBO |
|---|---|---|---|
| SLATE+ | **91.25** | 14.1 | 14.22 |
| LSD | 61.1 | 10.18 | 10.33 |
| Ours | 90.65 | **40.89** | **41.45** |

(d) Super-CLEVR

| Model | FG-ARI | mIoU | mBO |
|---|---|---|---|
| SLATE+ | 43.73 | 29.12 | 29.49 |
| LSD | 54.79 | 14.12 | 14.43 |
| Ours | **63.08** | **47.17** | **48.03** |

Figure 2: **Qualitative results on unsupervised object segmentation.** The baselines tend to split an object into different slots (CleverTex) and/or combine different objects and background into a single (MultiShapeNet, PGR, Super-CLEVR). On the other hand, our method produces consistently better object masks, showing improved disentanglement of objects and background in all datasets. More results are presented in the Figure 7. **Zoom in** for better view.

rand index for foreground objects (FG-ARI), mean intersection over union (mIoU), and mean best overlap (mBO). These metrics measure the overlap between the slot attention masks and ground-truth object masks, where FG-ARI focuses more on the coverage of theobject area.

## 5.1 UNSUPERVISED OBJECT SEGMENTATION

We first present the comparison results of our method with baselines on unsupervised object segmentation. Table 1 summarizes the quantitative results. Our method significantly improves the FG-ARI scores over the baselines in all datasets (8 to 29% improvement) except PTR, indicating that it captures an object holistically into an individual slot while the baselines tend to split the object into multiple parts and distribute it across multiple slots. In terms of mIoU and mBO, our method improves the baselines over all datasets, especially when the background is monolithic (MultiShapeNet, PTR, and Super-CLEVR). It indicates that the baselines struggle to separate the objects from the background when there exists a strong correlation between them, while our method can still robustly identify the objects. Overall, the results indicate that our method consistently outperforms the baselines by a significant margin. Notably, the consistent and significant improvement over LSD indicates that our regularization on the compositionality is effective in learning object-centric representation.

We also present the qualitative results in Figure 2. It shows that SLATE frequently splits the foreground object masks into multiple segments in CLEVRTex and Super-CLEVR datasets, and fails to capture object entities in PTR and MultiShapeNet. Similarly, LSD fails to segment the object in all datasets except CLEVRTex dataset, and tends to rely on positional bias in PTR and Super-CLEVR. In contrast, our method consistently captures objects with tight boundaries.

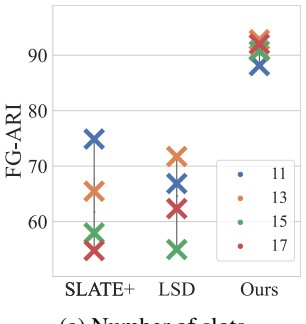 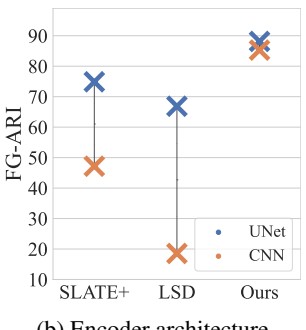 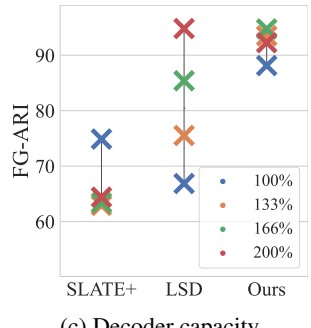

(a) Number of slots     (b) Encoder architecture     (c) Decoder capacity

Figure 3: **Robustness against various architectural biases.** We evaluate the robustness of our model various different number of slots, encoder architectures, and decoder capacities. Results based on mIoU and mBO are presented in Figure 5.

## 5.2 ROBUSTNESS OF COMPOSITIONAL OBJECTIVE

Compared to approaches based on auto-encoding, our method directly incorporates the objective to learn compositional representation, thus is more robust to the choice of architectural biases and hyperparameters. To demonstrate this, we evaluate our method while varying three major factors that are known to be highly sensitive in the previous approaches, such as number of slots, encoder architecture, and decoder capacity. Figure 3 summarizes the result on CLEVRTex dataset based on FG-ARI. All methods are trained up to 100K iterations for fair comparison.

**Number of slots**  Since object-centric learning assumes no prior knowledge on data, the mismatch between the number of objects and slots is inevitable in practice. To evaluate such robustness, we vary the number of slots from 11 to 17. Figure 3a presents the result. It shows that the performance of the baselines is highly sensitive to the number of slots. Specifically, SLATE tends to deteriorate more as the number of slots increases. Compared to the baseline, our method achieves more robust performance by encoding an object into a slot while leaving excess slots empty.

**Encoder architecture**  To identify the effect of slot encoder, we consider two popular architectures in the literature; a multi-layer CNN encoder (Singh et al., 2022b) and UNet-based encoder (Ronneberger et al., 2015). Figure 3b summarizes the result. It shows that employing the weaker encoder generally deteriorates the performance of the baselines significantly, indicating that architectural bias in the encoder is critical in the auto-encoding objective. Interestingly, the performance of our method is hardly affected by such drastic modifications, showing great robustness.

**Decoder capacity**  It is widely observed that the choice of decoder is also crucial in object-centric learning, since the highly expressive decoder can often bypass the object representation to minimize the reconstruction loss (Singh et al., 2022a). To examine such effect, we gradually increase the feature dimensions of the decoder to 133%, 166%, and 200%. Figure 3c summarizes the result. It shows that increasing the decoder capacity hampers the performance in SLATE. LSD exhibits the opposite trends showing a large improvement in FG-ARI, although its performance drops significantly in mIoU (Figure 5). Compared to the baselines, our method is much less sensitive to the decoder capacity, while the performance tends to improve slightly with increased capacity in all measures.

Overall, the results indicate that the quality of object-centric representation is significantly influenced by various factors in the auto-encoding-based methods. Conversely, our model consistently delivers outstanding performance across all configurations, even with major alterations to the encoder architecture. It demonstrates that our regularization through the composite path can directly encourage the model to learn compositional representation, greatly enhancing robustness to architectural biases.

## 5.3 INTERNAL ANALYSIS

**Component-wise Contributions**  To identify the contributions of each component in our framework, we conduct an ablation study and present the result in Table 2. The first row corresponds to our model with only the auto-encoding path, while the last row is the complete version of our model. Comparing the first row with the others shows that incorporating the composition path significantly improves overall quality. Adding $\mathcal{L}_{\text{Prior}}$, we observe a substantial improvement in all three metrics. Considering that FG-ARI measures the correct cluster membership of pixels within the objects,

Table 2: **Ablation study on CLEVRTex dataset.** All models are trained up to 100K iterations.

| $\mathcal{L}_{\text{Prior}}$ | $\mathcal{L}_{\text{Reg}}$ | Share $\mathbf{S}^{(0)}$ | FG-ARI | mIoU | mBO |
|:---:|:---:|:---:|:---:|:---:|:---:|
| ✗ | ✗ | ✗ | 42.48 | 52.26 | 52.41 |
| ✓ | ✗ | ✗ | 65.76 | 67.72 | 67.62 |
| ✓ | ✗ | ✓ | 70.29 | 69.08 | 69.28 |
| ✗ | ✓ | ✓ | 65.26 | 58.81 | 58.99 |
| ✓ | ✓ | ✓ | **88.15** | **75.30** | **75.64** |

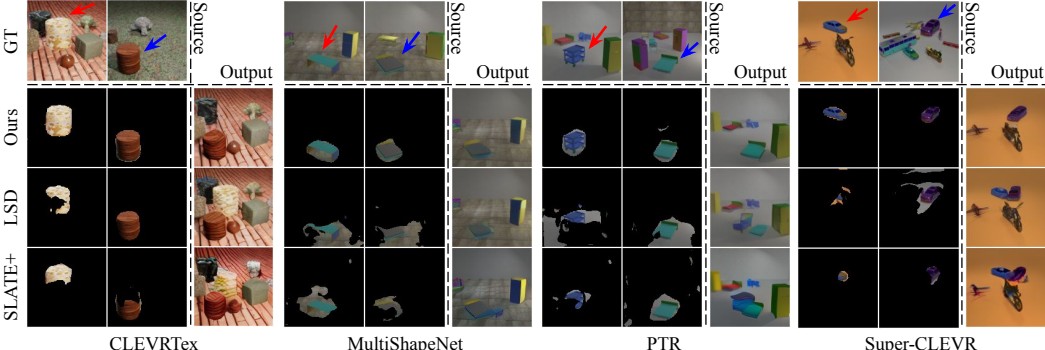

Figure 4: **Investigating object representation through compositional generation.** We investigate the compositionality of learned representations by removing (red arrow) and adding (blue arrow) object slots between two images and generating the composite image. More results are in Figure 8.

increased FG-ARI indicates that the generative prior encourages the encoder to capture more holistic object representations. This is because the generative prior penalizes the encoder for fragmenting the objects, thereby discouraging the generation of unrealistic partial objects in the composite image. Comparing the second and the third rows, we observe that sharing the slot initialization $\mathbf{S}^{(0)}$ slightly enhances mIoU and mBO scores. This improvement is likely attributed to the increased training stability by avoiding invalid slot combinations as shown in Figure 6. Incorporating regularization $\mathcal{L}_{\text{reg}}$ alone in the composition path does not improve the performance (fourth row), while combined with generative prior, it leads to significant improvement.

**Compositional Generation** We present the compositional generation results to further investigate the impact of our composition path. Figure 4 presents the results. Given two images, we construct the composite representation by replacing one object slot from the first image (red arrow) to another from the second image (blue arrow), and producing the image by the decoder. Based on visualization of the learned slots, we observe that the baselines often fail to learn compositional slot representation, by separating objects into multiple slots or encoding background with an object. It leads to failures in object-level manipulation, such as retaining an object after the removal (LSD in MultiShapeNet and PTR), altering the content of the added object (SLATE in MultiShapeNet), or transforming background with the object (SLATE in PTR and LSD in Super-CLEVR). In contrast, our method produces both semantically meaningful and realistic images from composite slot representations, supporting our claim that we can regularize object-centric learning through the proposed compositional path.

## 6 CONCLUSION

In this paper, we introduced a method to address the misalignment between object-centric learning and the auto-encoding objective. Our method is based on auto-encoding framework, and incorporates an additional branch to directly assess the compositionality of the representation. This involves constructing composite representations from two separate images and optimizing the encoder jointly with the auto-encoding path to maximize the likelihood of the composite image. Despite the simplicity, our extensive experiments demonstrate that our framework consistently improves the object-centric learning over the auto-encoding frameworks. It also shows that our method greatly enhances the robustness to the choice of architectural biases and hyperparameters, which typically pose sensitivity challenges in auto-encoding-centric approaches.

**Acknowledgements** This work was supported in part by Institute of Information & communications Technology Planning & Evaluation (IITP) grant (No.2022-0-00926, 2022-0-00959, 2021-0-02068, and 2019-0-00075) and National Research Foundation of Korea(NRF) grant (2021R1C1C1012540 and 2022R1C1C1009443) funded by the Korea government(MSIT).

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

# A    ADDITIONAL IMPLEMENTATION DETAILS

Table 3 provides details of hyperparameters used in experiments. For the Slot Attention encoder $E_\theta$ and a diffusion decoder $D_\phi$, we base our implementation on Jiang et al. (2023). Specifically, in the Slot Attention encoder, we employ a CNN-based UNet image encoder. Prior to the UNet encoder, we incorporate a single layer CNN to downsample the original $128 \times 128$ image to a $64 \times 64$ image. Implementing the diffusion decoder $D_\phi$, we follow the design of the LSD decoder. The overall structure of $D_\phi$ is based on the U-Net architecture, where each layer is composed of CNN layers and a transformer layer. The surrogate decoder $D_\psi$ is implemented with the Transformer Architecture in Singh et al. (2022a). It takes slots as input through cross-attention layers. In the experimental setting, we augment the Super-CLEVR dataset by randomly altering the background color to another color.

| General | Batch Size | 64 |
|---|---|---|
| | Training Steps | 200K |
| | Learning Rate | 0.0001 |
| CNN Backbone | Input Resolution | 128 |
| | Output Resolution | 64 |
| | Self Attention | Middle Layer |
| | Base Channels | 128 |
| | Channel Multipliers | [1,1,2,4] |
| | # Heads | 8 |
| | # Res Blocks / Layer | 2 |
| | Slot Size | 192 |
| Slot Attention | Input Resolution | 64 |
| | # Iterations | 7 |
| | Slot Size | 192 |
| Auto-Encoder | Model | KL-8 |
| | Input Resolution | 128 |
| | Output Resolution | 16 |
| | Output Channels | 4 |
| Diffusion Decoder | Input Resolution | 16 |
| | Input Channels | 4 |
| | $\beta$ scheduler | Linear |
| | Mid Layer Attention | Yes |
| | # Res Blocks / Layer | 2 |
| | # Heads | 8 |
| | Base Channels | 192 |
| | Attention Resolution | [1,2,4,4] |
| | Channel Multipliers | [1,2,4,4] |
| Surrogate Decoder | Layers | 8 |
| | # Heads | 8 |
| | Hidden Dim | 384 |

Table 3: **Hyperparameters used in our experiments.**

# B    ADDITIONAL RESULTS

## B.1    ADDITIONAL RESULTS ON ROBUSTNESS TESTS

We include results of the robustness test on mIoU, mBO metrics in Figure 5. Similar to the results on FG-ARI (Figure 3), our model is surprisingly robust to a wide range of hyperparameters. It suggests that directly optimizing the compositionality of the representation significantly reduce a dependency on a choice of hyperparameters.

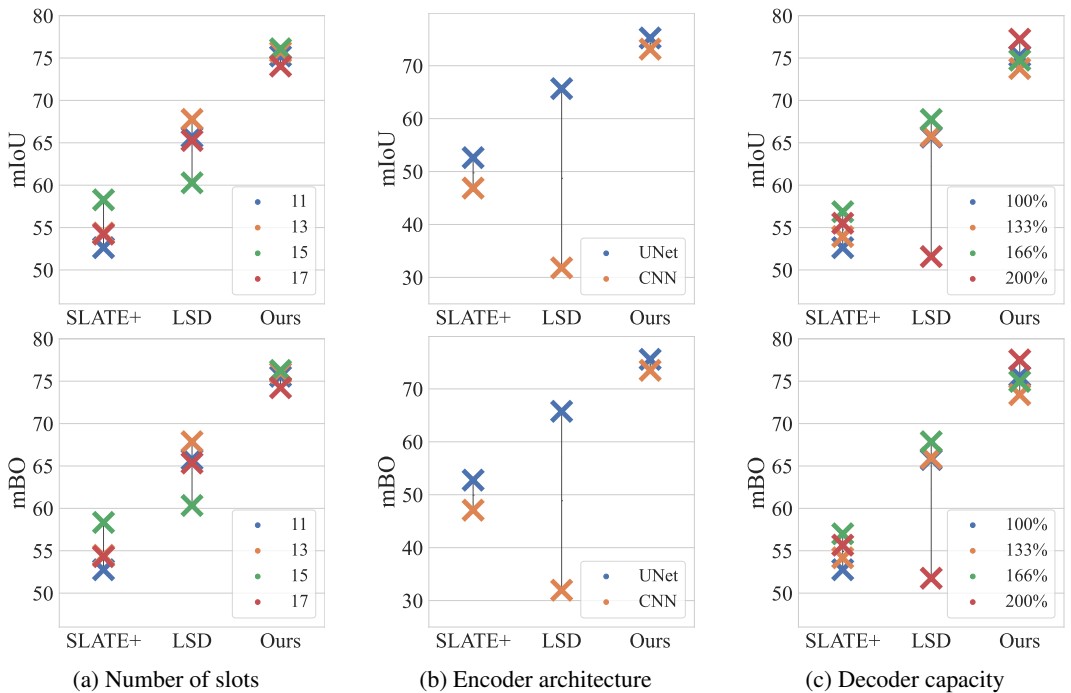

Figure 5: **Robustness across Various Hyperparameters.** We evaluate the robustness of our model across different number of slots, encoder type, and decoder capacity. Among various hyperparameters, our model steadily shows powerful performance against baselines.

### B.2 Unsupervised Object Segmentation

We present additional qualitative results for unsupervised segmentation results in Figure 7. Our method successfully segmented the object regions across all four datasets. In contrast, baselines easily divide each object into multiple segments or capture a wide area around the objects.

### B.3 Effect of Mixing Slot Strategy

As discussed in Section 3.1 and Section 5.3, sharing $\mathbf{S}^{(0)}$ slightly enhances the performance by roughly avoiding suspicious compositions during training. To investigate how sharing slot initialization affects the composition, we obtained the slot representations from multiple scenes with the same slot initialization and grouped those representations by their order, *i.e.*, $\mathbf{s}_i$ belongs to $i$-th group. Figure 6, we observe that the captured objects from the same initialization are correlated to some degree. The slots in the first row mostly capture the backgrounds of the scenes, while other slots tend to capture foreground objects. Moreover, we observe that the slots in the fourth row tend to capture the objects located in the lower part of the scene. Based on these observations, we conjecture that sharing slot initialization stabilizes our framework by alleviating some suspicious compositions, such as the occlusion of foreground objects or composing multiple backgrounds.

### B.4 Investigation on Compositionality of Slots

In this section, we provide more visual samples of composite images to investigate the compositionality of slot representations in our method. Figure 8 illustrates the results of generating composite images by mixing slots from two images, which supplements the Figure 4 in the main paper. It shows that the baselines often fail to capture compositional objects into independent slots, while our method successfully learns object-level slots through the composition path. As a result, the composite images generated by the baselines often fail to adhere the object-level manipulation, such as retaining the removed objects or transforming the object identity and background pattern while adding a new object. In contrast, our method preserves these semantics more precisely based on accurate object slots.

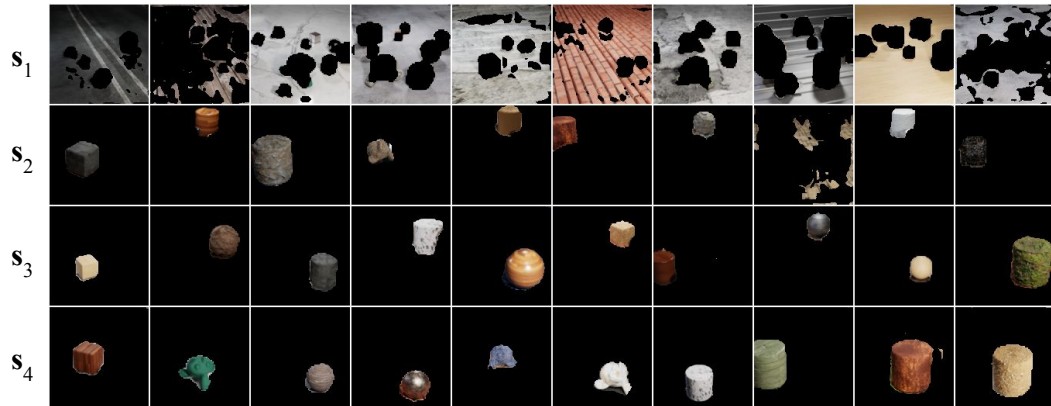

Figure 6: **Grouping of slots by sharing the slot initialization.** We obtain slot representation from various images while sharing the initial values of $\mathbf{S}^{(0)}$ and cluster the representation based on their initial values. Slots initialized as $\mathbf{s}_1$ consistently capture backgrounds.

## B.5 ADDITIONAL QUALITATIVE RESULTS ON COMPOSITIONAL GENERATION

To help a comprehensive understanding of the baselines, we provide more qualitative samples on compositional generation in Figure 9. While Figure 4 and Figure 8 illustrates the common failure cases of the baselines, we additionally present compositional generation results where the baselines also reasonably capture an object into a slot. Despite the reasonable slot attention masks, the composite image produced by the baseline model often distorts the original appearance of the object or creates unrealistic partial objects. In contrast, our model consistently produces faithful composite images, which highlights the importance of the compositional objective.

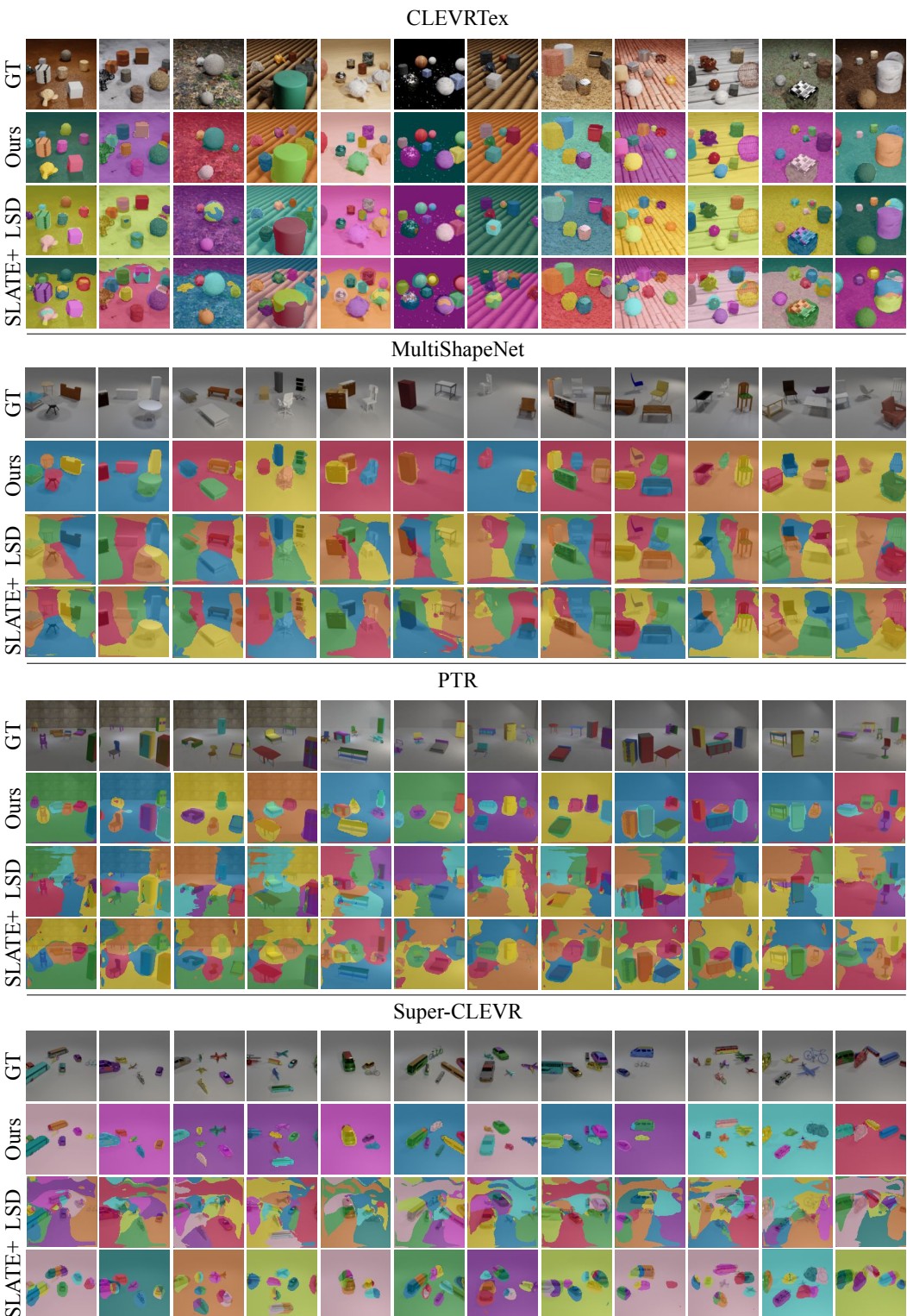

Figure 7: **More Qualitative Results on Unsupervised Object Segmentation.**

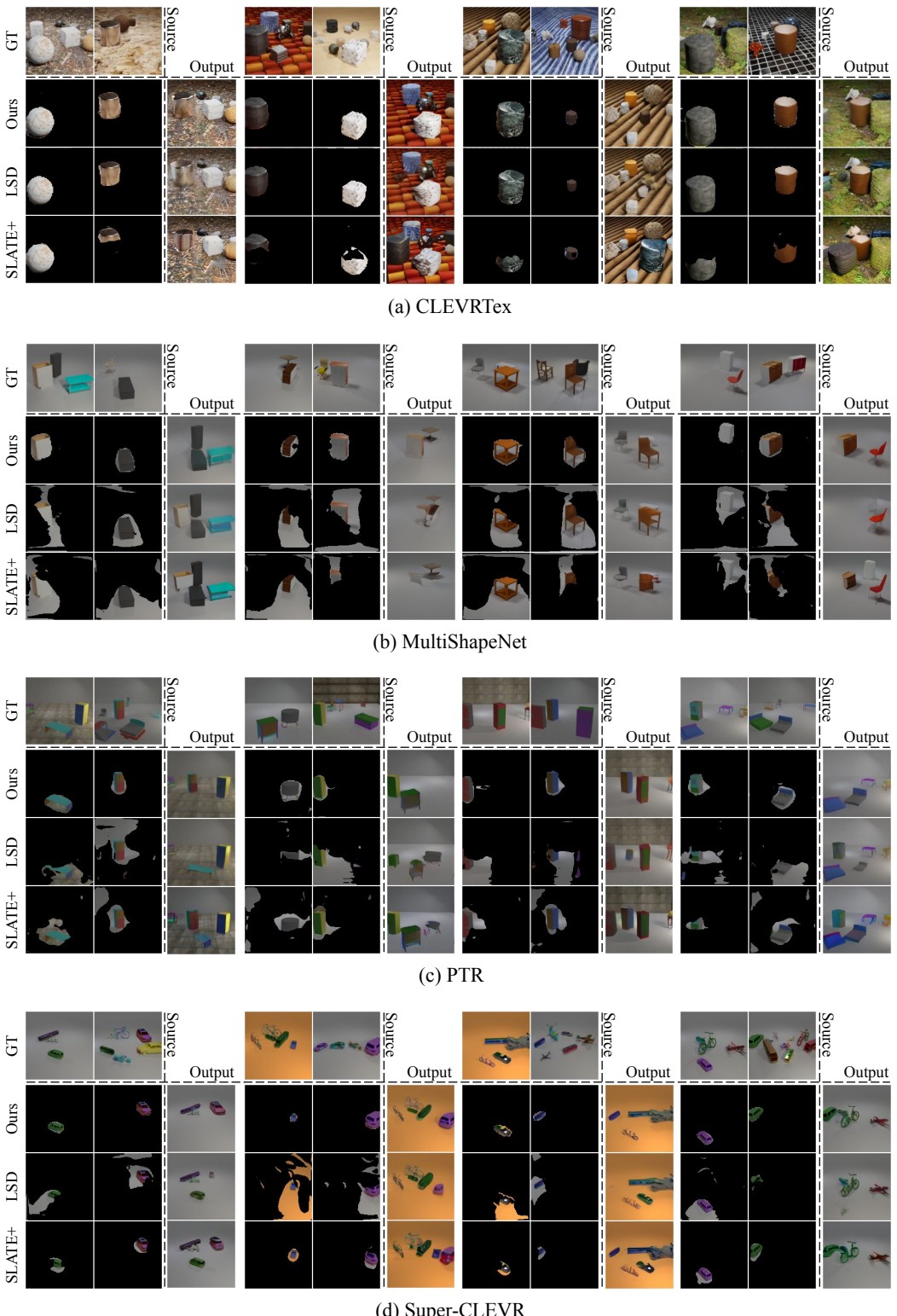

Figure 8: **More Qualitative Results on Compositional Generation between two images.**

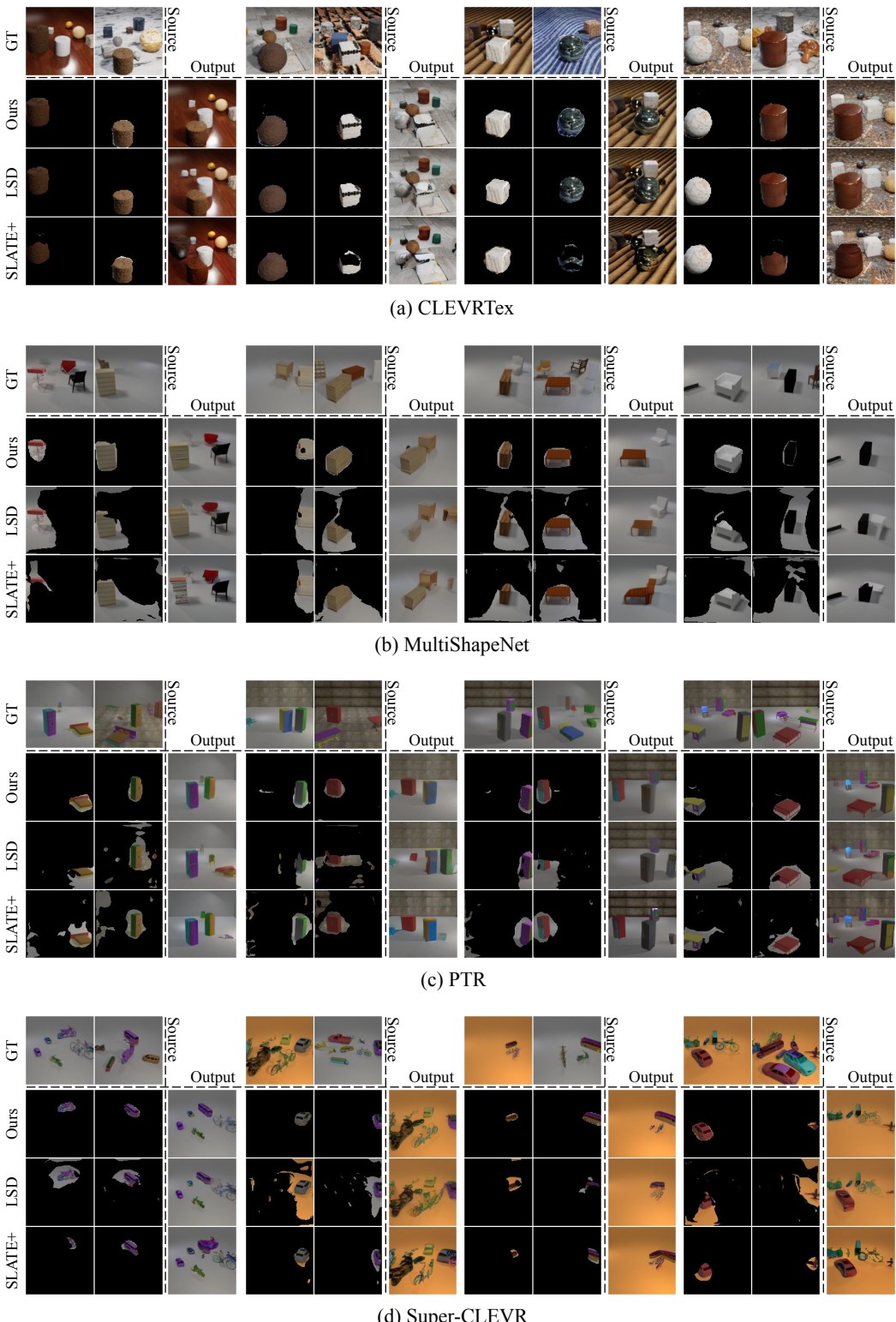

Figure 9: **More Qualitative Results on Compositional Generation between two images.**

Table 4: **Results on object property prediction.** We evaluate the quality of the learned representation through object property prediction. Our model consistently performs better than the baselines across different properties and datasets.

| Dataset | CLEVRTex | | | PTR | | | Super-CLEVR | | |
|---|---|---|---|---|---|---|---|---|---|
| Property | Position ($\downarrow$) | Shape ($\uparrow$) | Material ($\uparrow$) | Position ($\downarrow$) | Shape ($\uparrow$) | Position ($\downarrow$) | Shape ($\uparrow$) | Material ($\uparrow$) |
| SLATE+ | 0.1757 | 78.72 | 67.99 | 0.2218 | 88.21 | 0.5397 | 76.28 | 68.43 |
| LSD | 0.1563 | 85.07 | 82.33 | 0.5999 | 75.80 | 0.4372 | 76.5 | 69.24 |
| Ours | **0.1044** | **88.86** | **84.29** | **0.1424** | **90.00** | **0.4262** | **80.67** | **71.31** |

## B.6 ADDITIONAL EVALUATION ON OBJECT PROPERTY PREDICTION

To assess the quality of acquired object representations, we employ object property prediction using the learned representation, following the methodology outlined in Jiang et al. (2023); Dittadi et al. (2022). During this process, we train a network to predict the property based on a fixed slot representation. The true label for the slot representation is established through Hungarian matching, comparing the mask of slots with the foreground objects. The remaining slots after matching are considered as backgrounds. For predicting properties, we employ a 4-layer MLPs with a hidden dimension of 196. Accuracy is reported for categorical properties, while mean squared error is reported for continuous properties. We assess the models on datasets that include object properties.

The results for object property prediction are presented in Table 4. Our model consistently performs better than the baselines across different properties and datasets. Notably, it excels in predicting shape and position, as observed in the high segmentation performance depicted in Figure 2 and Table 1. Furthermore, our model demonstrates improved performance in predicting materials indicating its ability to capture local and high-frequency information.

On the Super-CLEVR dataset, despite our model's higher segmentation performance, the mean square error of position remains competitive with other baselines. We attribute this to the challenging nature of the dataset, where scenes often include many small and occluded objects. As a result, both our model and the baselines face increased difficulty in predicting position, leading to a higher error rate compared to other datasets.

## B.7 Additional Results on Real-world dataset

To explore the scalability of our novel objective in a complex real-world dataset, we examine our framework in BDD100k dataset Yu et al. (2020), which consists of diverse driving scenes. Since the images captured on night or rainy days often produce blurry and dark images, we filter the data to collect only sunny and daytime images using metadata, which leaves about 12k, 1.7k images in the training/validation set, respectively. Since it has been widely observed that learning the object-centric representation directly on real-world dataset is challenging, we bootstrap our auto-encoding path with off-the-shelf models following Jiang et al. (2023). Specifically, we employ pretrained DINOv2 Oquab et al. (2023) and Stable Diffusion Rombach et al. (2022) for the image encoder and slot decoder in our auto-encoding path, respectively. Instead of using frozen Stable-Diffusion, we update key and value mapping layers in cross-attention layers to enhance the overall auto-encoding performance following Kumari et al. (2023). For efficient training, we first warm up the auto-encoding path for 200k iterations and then train only the surrogate decoder for 140k iterations on top of frozen slot representations, which significantly boosts up the convergence of the surrogate decoder. Finally, we optimize our compositional path for 100k iterations. For the baseline, we compare our model trained with only auto-encoding objective for 300k iterations, which converges closely to the Stable-LSD Jiang et al. (2023).

Figure 10 illustrates qualitative results on unsupervised object segmentation. The slot attention masks of our model successfully capture composable instances such as cars, buildings, trees, font hoods, etc. In contrast, the diffusion model trained without compositional objective often divides the objects into multiple slots or encodes multiple objects into a slot. For example, the car or truck is frequently divided into multiple masks, and multiple cars are often encoded into a single slot.

To further examine the compositionality of the learned slot representations, we qualitatively analyze the visual samples of composite images in Figure 11 similar to Section B.4. We observe that our method successfully generates realistic scenes, modeling complex correlations among objects and environments. It appropriately adapts the appearance of newly added/removed objects, their shadow, reflections in the front glass and hood, and sometimes even global illumination change caused by removing the sun. In contrast, the auto-encoding model often fails to achieve faithful composition. For example, in Row 1 of Figure 11, the car still appears in the composite image even after the removal of the corresponding slot. Also, we observe that removing slots containing partial information of the object often leads to undesirable artifacts in composite images such as creating a new car in the first example of Row 2, or leaving unrealistic artifacts in the third example of Row 2. In contrast, our model produces natural object-wise manipulation. Moreover, the baseline model often fails to faithfully generate the inserted object as shown in Row 3, while our model tends to maintain the target object. In Row 4, we identify that our model successfully models complex interaction between slots such as removing sunlight changing the reflection of the bonnet in the first image, or changing a blurry car into a sharp car corresponding to bright weather. In summary, we identify that our novel objective on compositionality can help to learn object-wise disentanglement even in complex scenes and helps to model complex interactions among objects.

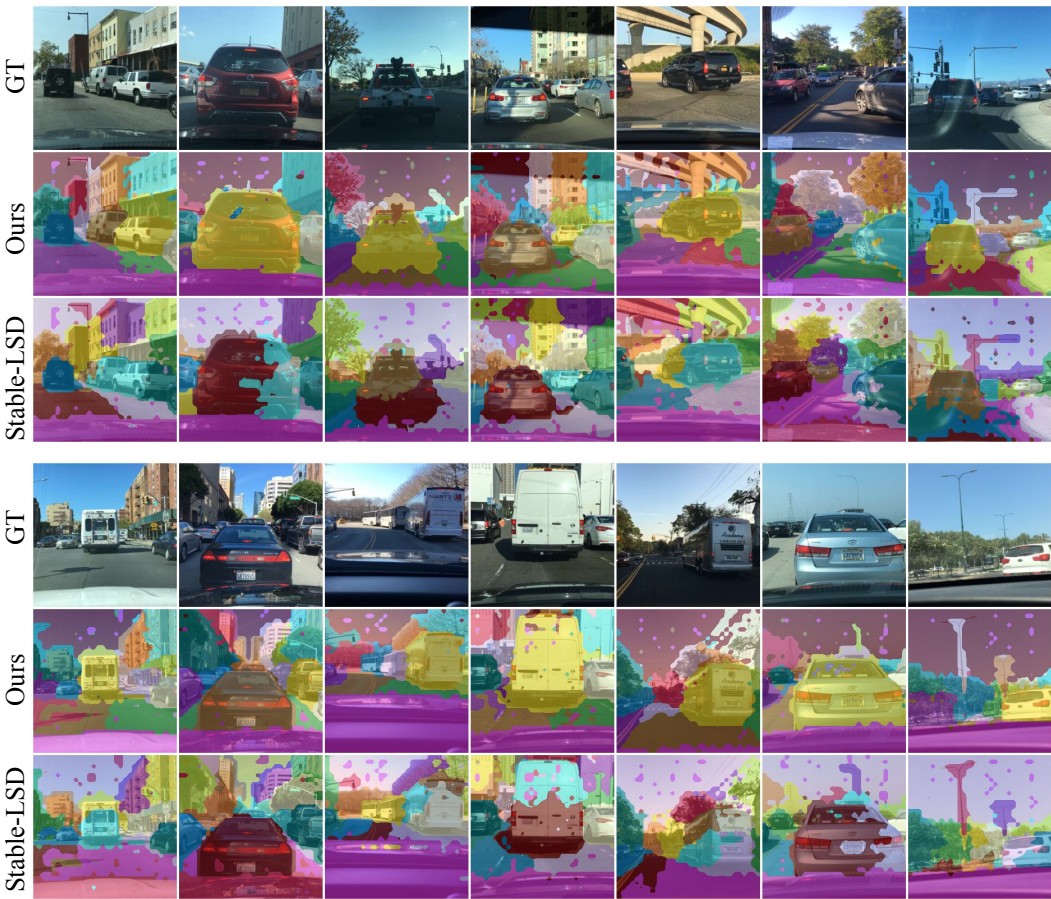

Figure 10: **Qualitative results on unsupervised segmentation in BDD100k.**

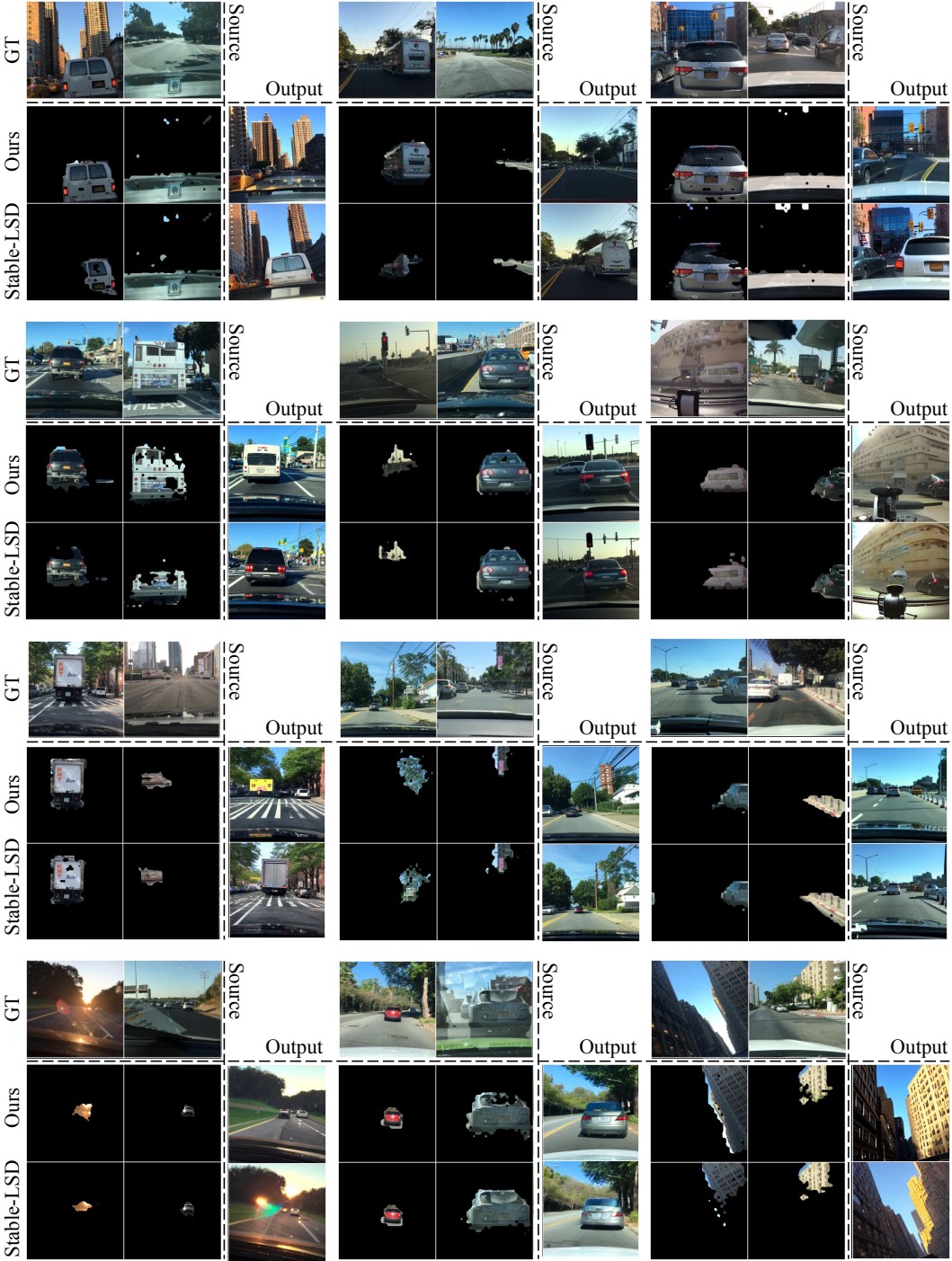

Figure 11: **Qualitative results on compositional generation in BDD100k.**

