# OpenReview forum: "Learning to Compose: Improving Object Centric Learning by Injecting Compositionality"
_ICLR.cc/2024/Conference — ICLR 2024 poster_

### Official Review · Reviewer_H2Bo · 2023-10-31

**Soundness:** 3 good
**Presentation:** 3 good
**Contribution:** 2 fair
**Rating:** 6
**Confidence:** 4

**Summary:**

This paper focuses on the compositional ability in object-centric learning. Existing works mostly rely on auto-encoding training paradigm and may sacrifice the object disentanglement. While this work explicitly introduces an object composition path in addition to the original reconstruction objective, and employs generative prior to validate the rendered object compositions. The experiments show that the proposed method achieves better resutls in object disentanglement and enhances the robustness to hyper-parameters.

**Strengths:**

1. The paper is well motivated. Compositionality is a vital property in object-centric perception and generation. The authors conclude the potential weakness in existing auto-encoding based object-centric learning methods and introduce an object composition path for explicit optimization on composition.
2. The method is simple and intuitive. It uses generative prior to help validate the rendered object composition from mixed slot representations and guides the object disentanglement and composition learning.
3. The experiments show the improvement on object-centric benchmarks. And the proposed method is robust to the number of slots, which is a significant hyper-parameter in object-centric learning and difficult to determine.

**Weaknesses:**

1. Most experiments are conducted on the synthetic objects, more experiments on realistic complex scenes, e.g., COCO, are desired.
2. More compared methods should be included, e.g., DINOSAURE [1].
3. There are some recent works focusing on the binding ability of the slots to specific object types or properties, e.g., [2]. Is it possible to employ the binding ability to enhance the interpretablity of slot mixing and further enahnce the composition ability?

[1] Seitzer et al. Bridging the gap to real-world object-centric learning. ICLR 2023.
[2] Jia et al. Improving Object-centric Learning with Query Optimization. ICLR 2023.

**Questions:**

See weakness

---

> ### Author Response · Authors · 2023-11-22
> **Official Response to Reviewer H2Bo (1/2)**
>
> We thank the reviewer for the positive comments. Below we respond to the individual questions.
>
> > **Q1.**  Most experiments are conducted on synthetic objects, more experiments on realistic complex scenes, e.g., COCO, are desired.
>
> **A1.** We appreciate the valuable comments. First, we would like to kindly note that learning object-centric representation in complex real-world images is widely perceived as challenging, and existing works rely on frozen pre-trained encoders and decoders to alleviate the problem. Since freezing the image encoder with a shallow slot attention module limits the opportunity to investigate the true impact of the compositional objective, we follow the standard settings that employ the synthetic data for evaluation.
>
> Nonetheless, we agree with the reviewer that the synthetic scenes are insufficient to demonstrate the scalability of our compositional objective to complex scenes. Following the reviewer’s suggestion, we conducted preliminary experiments on the BDD100K dataset, which contains real-world driving scenes with many objects (cars, signs, buildings, trees, pedestrians, etc.) as well as complex interactions among objects and with the environment (occlusion, shadow, lighting, reflection, etc.). Following the prior work [1], we employed the pre-trained DINOv2 [2] as an image encoder and Stable Diffusion [3] as the generative prior.  We then apply warm-up training for the one-shot decoder for 140K steps with an auto-encoding objective, and apply our compositional objective for the next 100K steps.
>
> The results are summarized in Section B.7 in the Appendix (Figures 10 and 11). Based on visualization of the slot attention map (Figure 10), we observe that our method captures composable object instances not only from the foreground such as cars, buses, and trucks, but also from static environments such as road signs, trees, buildings, sidewalks, etc. In contrast, learning with only diffusion loss often separates an object into multiple slots, or contains multiple objects into a single slot. When composing two images by mixing their slots (Figure 11), we observe that our method successfully generates realistic scenes, modeling complex correlations among objects and environments. It appropriately adapts the appearance of newly added/removed objects, their shadow, reflections in the front glass and hood, and sometimes even global illumination change caused by removing the sun. Our method also generates more accurate compositions compared to the baseline, which often fails to adhere to the addition or removal of the objects.
>
> To summarize, our preliminary experiments showed promising results that the proposed compositional objective can scale to complex scenes and model complex correlations among objects. We believe that incorporating the compositional objective in self-supervised learning can greatly enhance object-centric learning in complex scenes through end-to-end training with the deep encoder, which we plan to investigate in future work.
>
> [1] Jiang et al., Object-centric slot diffusion, NeurIPS 2023.
>
> [2] Oquab et al., DINOv2: Learning robust visual features without supervision, arXiv preprint 2023.
>
> [3] Rombach et al., High resolution image synthesis with latent diffusion models, CVPR 2022.

---

> ### Author Response · Authors · 2023-11-22
> **Official Response to Reviewer H2Bo (2/2)**
>
> > **Q2.**  Comparison to DINOSAUR.
>
> **A2.** We appreciate the comment. As suggested by the reviewer, we included the DINOSAUR as an additional baseline and reported the comparison results in the table below. Although it performs reasonably well in CLEVRTex and PTR datasets, it fails dramatically in MutiShapeNet and Super-CLEVR datasets; we observe that it is because DINOSAUR encodes the entire foreground objects into a single slot in these datasets. Such failures are attributed to the frozen image encoder, which fails to generalize to domains unseen during the pre-training. It shows that learning the image encoder end-to-end with object-centric representation is important for building a robust and general object-centric learning framework.
>
> We would also like to note that the main contribution of the DINOSAUR is employing the pre-trained features in object-centric learning, which is orthogonal to our contribution and hence can be combined together. In Section B.7 in the Appendix, we provided results on a real-world dataset, where we incorporated the compositional loss on top of the pre-trained DINO.
>
>
> | **CLVERTex** | **FG-ARI (↑)** | **mIoU (↑)** | **mBO (↑)** |
> |--------------|:--------------:|:------------:|:-----------:|
> |   DINOSAUR   |      70.67     |     48.20     |    50.03    |
> |    SLATE+    |      71.29     |     52.04    |    52.17    |
> |      LSD     |      76.44     |     72.32    |    72.44    |
> |     Ours     |    **93.06**   |   **74.82**  |  **75.36**  |
>
>
> | **MultiShapeNet** | **FG-ARI (↑)** | **mIoU (↑)** | **mBO (↑)** |
> |-------------------|:--------------:|:------------:|:-----------:|
> |      DINOSAUR     |      1.99      |     12.53    |     20.30    |
> |       SLATE+      |      70.44     |     15.55    |    15.64    |
> |        LSD        |      67.72     |     15.39    |    15.46    |
> |        Ours       |    **89.80**    |   **59.21**  |   **59.40**  |
>
> | **PTR**  | **FG-ARI (↑)** | **mIoU (↑)** | **mBO (↑)** |
> |----------|:--------------:|:------------:|:-----------:|
> | DINOSAUR |      58.23     |     23.20     |    26.48    |
> |  SLATE+  |    **91.25**   |     14.10     |    14.22    |
> |    LSD   |      61.10      |     10.18    |    10.33    |
> |   Ours   |      90.65     |   **40.89**  |  **41.45**  |
>
>
> | **SuperCLEVR** | **FG-ARI (↑)** | **mIoU (↑)** | **mBO (↑)** |
> |----------------|:--------------:|:------------:|:-----------:|
> |    DINOSAUR    |      2.92      |      5.70     |    8.93    |
> |     SLATE+     |      43.73     |     29.12    |    29.49    |
> |       LSD      |      54.79     |     14.12    |    14.43    |
> |      Ours      |    **63.08**   |   **47.17**  |  **48.03**  |
>
> ---
> > **Q3.**  There are some recent works focusing on the binding ability of the slots to specific object types or properties, *e.g.*, “ Improving Object-centric Learning with Query Optimization”. Is it possible to employ the binding ability to enhance the interpretability of slot mixing and further enhance the composition ability?
>
> **A3.** We appreciate the insightful comment. Indeed, our work investigates a general problem of learning compositional representation and employs simple mixing strategies for generality.  We believe that incorporating the binding ability of the slots can enhance the mixing strategy by reducing the search space for the valid slot combination, and improving the interpretability of the composite representation which is useful especially in manipulation tasks.

---

### Official Review · Reviewer_f9fX · 2023-10-31

**Soundness:** 2 fair
**Presentation:** 3 good
**Contribution:** 3 good
**Rating:** 6
**Confidence:** 4

**Summary:**

The paper introduces a generative objective to regularize the learned representation of a Slot Attention model. In particular, the generative objective ensures that decoding a composition of slots from different images results in a realistic image. The additional regularizer trades off reconstruction quality for better compositionality of the learned representation.

**Strengths:**

The authors identify a weakness in the standard reconstruction objective of Slot Attention and similar papers: the reconstruction objective does not explicitly encourage compositionality of the learned representation. This observation calls for a generative objective that ensures that composition of slots can be decoded into realistic images. Instead of implementing this with a GAN discriminator, which would require an additional component, the authors follow the approach of Poole et al. (2022) where the diffusion model is reused as a generative prior.

The idea of using a cheap surrogate one-shot decoder to approximate an expensive denoising process is quite clever and surely helps speeding up training.

The qualitative results on compositional generation are convincing. I would like to see if the method would work on real-world images like the ones in COCO.

**Weaknesses:**

In the proposed method, the number of losses, regularizers, and training tricks to balance is high. I would appreciate a more thorough ablation study where each component is isolated and tested. The combinations in table 2 are valid but do not cover all possibilities.

I would appreciate additional evaluation metrics that are not based on segmentation. Learning a good object-centric representation means much more than achieving good segmentation. It is important to show that the learned slots are useful for downstream tasks such as counting, tracking, property prediction, etc. This way the claim made in the introduction that "[the method] significantly boosts the overall quality of object-centric representations" can be supported. My rating on "soundness" would be higher if the authors had included such experiments.

**Questions:**

May I recommend a bar plot for Figure 3? The current plot based on colored Xs is quite hard to read. Moreover, a bar plot would allow to show the standard deviation of the results over multiple independent runs, which is important to strengthen the claims made in the text.

Can you provide non-segmentation based evaluation metrics? They can be the same as in Jiang et al. (2023) for example.

In Jiang et al. (2023) the compositional generation appear to be much stronger that what is shown in Figure 4. It seems to me that for the proposed method the authors chose a source image where all objects are well separated and therefore the composition looks good. Whereas the segmentation in LSD and SLATE+ on the same image is poor and therefore the composed image looks bad. Can you clarify in the caption if the examples were cherry-picked? Can you find another image where all 3 methods segment correctly the objects that you want to compose? This way the comparison would be fairer.

Typo: "is not contrained to"

---

> ### Author Response · Authors · 2023-11-22
> **Official Response to Reviewer f9fX (1/2)**
>
> We thank the reviewer for the positive comments. Below we respond to the individual questions.
>
> > **Q1.** In the proposed method, the number of losses, regularizers, and training tricks to balance is high. I would appreciate a more thorough ablation study where each component is isolated and tested. The combinations in table 2 are valid but do not cover all possibilities.
>
> **A1.** We appreciate the valuable comment. Following the reviewer’s suggestion, we report the ablation study on all valid combinations of the components in the table below. We observe that the overall trends remain the same: considering the composition of two images for object-centric learning, either via composition loss or regularization, is helpful, and adding the slot initialization also improves the performance as well as enabling stable training and faster convergence. Overall, all components of our method contribute to the improvement, while employing composition loss and regularization are the most critical. We appreciate the comment and will include the results in the paper.
>
> | **L_prior** | **L_Reg** | **Share slot init** | **FG-ARI** |  **mIoU** |  **mBO**  |
> |:-----------:|:---------:|:-------------------:|:----------:|:---------:|:---------:|
> |      X      |     X     |          X          |    42.48   |   52.26   |   52.41   |
> |      ✔      |     X     |          X          |    65.76   |   67.62   |   67.62   |
> |      ✔      |     X     |          ✔          |    70.29   |   69.08   |   69.28   |
> |      X      |     ✔     |          ✔          |    65.26   |   58.81   |   58.99   |
> |      X      |     ✔     |          X          |    57.58   |    53.50   |    53.70   |
> |      ✔      |     ✔     |          X          |  **88.72** |   73.24   |   73.63   |
> |      ✔      |     ✔     |          ✔          |    88.15   | **75.30** | **75.64** |
>
> ---
>
> >**Q2.** Can you provide non-segmentation based evaluation metrics? They can be the same as in Jiang et al. (2023) for example.
>
> **A2.** We appreciate the valuable comment. Following the reviewer’s suggestion, we follow the evaluation protocol of Jiang et al. (2023) and evaluate the quality of the learned representation by predicting the object properties from the slots using object-wise annotations in CLEVRTex, PTR, and Super-CLEVR datasets (we could not include the MultiShapeNet dataset due to missing object labels). We report the results in Section B.6 in the Appendix and the tables below.
>
> Overall, we observe that our method consistently outperforms the baselines in all datasets and properties, showing that the learned representations capture the characteristics of the objects well. We found that the prediction error in Super-CLEVR dataset tends to be higher than the other datasets, since it contains many small and occluded objects. We appreciate the comment and will include the results in the paper.
>
> | **CLEVRTex** | **SLATE+** | **LSD** |  **Ours**  |
> |:------------:|:----------:|:-------:|:----------:|
> | Position (↓) |   0.18   |  0.16 | **0.10** |
> |   Shape (↑)  |    78.72   |  85.07  |  **88.86** |
> | Material (↑) |    67.99   |  82.33  |  **84.29** |
>
>
> |    **PTR**   | **SLATE+** | **LSD** |  **Ours**  |
> |:------------:|:----------:|:-------:|:----------:|
> | Position (↓) |   0.22   |  0.60 | **0.14** |
> | Shape (↑) |    88.21   |   75.80  |   **90.00**   |
>
>
> | **Super-CLEVR** | **SLATE+** | **LSD** |  **Ours**  |
> |:--------------:|:----------:|:-------:|:----------:|
> |  Position (↓)  |   0.54   |  0.44 | **0.43** |
> |    Shape (↑)   |    76.28   |   76.50  |  **80.67** |
> |  Material (↑)  |    68.43   |  69.24  |  **71.31** |

---

> ### Author Response · Authors · 2023-11-22
> **Official Response to Reviewer f9fX (2/2)**
>
> > **Q3.** May I recommend a bar plot for Figure 3? The current plot based on colored Xs is quite hard to read. Moreover, a bar plot would allow to show the standard deviation of the results over multiple independent runs, which is important to strengthen the claims made in the text.
>
> **A3.** We appreciate the comment. We will replace Figure 3 with a bar or box plot to present the standard deviations of each run more clearly.
>
> ---
>
> > **Q4.**  In composition generation (Figure 4), please specify whether the outputs are cherry-picked. If so, please provide cases where all three methods reasonably work for fair comparison.
>
> **A4.** Figure 4 illustrates the common failure cases of the baselines, which are caused by encoding an object into multiple slots (*e.g.*, separating an object into multiple parts). For a more comprehensive understanding of the baselines, we also present additional results in Section B.5 (Figure 9) in the Appendix. In this figure, we present composite generation results similar to Figure 4 but for the examples where the baselines capture an object into a slot successfully. It shows that despite the reasonable slot attention map, the composite images produced by the baselines often exhibit unsatisfactory quality by distorting objects or altering their appearances due to the addition or removal of the objects. It shows that the object-centric representation learned in the baselines is generally not as compositional as our method.

---

### Official Review · Reviewer_6966 · 2023-11-01

**Soundness:** 4 excellent
**Presentation:** 3 good
**Contribution:** 3 good
**Rating:** 8
**Confidence:** 4

**Summary:**

The authors propose a scheme for encouraging compositionality in slot-based Object-Centric Learning (OCL) models by introducing a compositional objective alongside the standard auto-encoding reconstruction loss. In particular, they enforce that mixing subsets of slots from two separate scenes should result in a "valid" image, as determined by a generative prior which is trained through the auto-encoding process - this is accomplished by using a diffusion model as the decoder. To enable fast-decoding and cheaper evaluation of gradients for the encoder through the compositional path, they also train a transformer-decoder on the autoencoding tasks.

They perform extensive experiments and ablations, demonstrating the efficacy of their method against other state-of-the art methods, both in terms of performance and robustness to hyperparameters such as slot-count and decoder size (which are sensitive in many OCL methods). To the best of my knowledge, the idea of encouraging compositionality of slots through a valid-mixing objective is novel and forms a valuable contribution to the OCL community.

**Strengths:**

The paper's methodology is well described with helpful figures, and the experiments are comprehensive. Notably, the experiments on parameter robustness in sec 5.2, as well as the qualitative results shown alongside experiments with slot mixing strategy in the appendices, provide a compelling case for the benefits of this method against other SoTA methods, beyond the mere improvement of performance.

**Weaknesses:**

There are no major weaknesses with the methodology or evaluations. The presentation is clear in most places, but there are many grammatical mistakes. It is recommended that these be fixed through the use of automated grammar checkers or proof-reading by a native speaker.

Some minor notes on clarification:
* On the first reading, it was not entirely clear how the one-shot decoder and diffusion decoders were being trained (i.e. whether one, or both, were being used in the auto-encoding path) - this is made clear at the start of sec 3.3 "in auto-encoding path [...] two different decoders [...] are trained", but it would be good to make it more explicit in prior paragraphs when the relevant decoders are introduced and their training is discussed.
* Related to the above, it would be good to make it clear in the figure that *both* decoders are trained/used on the auto-encoding branches, for example by putting "(both) Decoder" on the branches

**Questions:**

Overall the work is very interesting and comprehensively explained, and I was left only with minor questions:
1) *Why background slots do not conflict*: Could the authors explain whether / why, if not, background slots conflict when sampling slots with the random mixing strategy? In this case 1) is there any reason to expect canonical slot ordering, such that two background slots are unlikely to be sampled or 2) any reason why if two background slots are sampled, this does not lead to invalid scenes?
2) *On the relative contributions of Regularization+Shared Slot Initializations (Row 2) and the compositional prior (Row 1) in Table 2*: It is Interesting that Row 2, is roughly as effective as Row 1. Was there a significant difference in the qualitative behaviour of the models in these two settings; i.e. did Row 1 yield "better" slots, but lead to worse metrics for some other reason - or can we conclude that the bias towards compositional slot representations in both of these settings was comparably strong? (which would be somewhat surprising!). Perhaps the decoder receives a stronger learning signal in Row 2, and in Row 1 the generative prior acting by itself is constrained because the decoders don't learn as well, and so provide a weaker compositional signal?

---

> ### Author Response · Authors · 2023-11-22
> **Official Response to Reviewer 6966**
>
> We thank the reviewer for the positive comments. Below we respond to the individual questions.
>
> >**Q1.** Suggestions to improve presentation
>
> **A1.** We appreciate the valuable comment. We will modify Figure 1 and the first paragraph of the Section 3 to clarify that we are learning two decoders simultaneously.
>
> ---
> >**Q2.**  Why background slots do not conflict: Could the authors explain whether / why, if not, background slots conflict when sampling slots with the random mixing strategy? In this case 1) is there any reason to expect canonical slot ordering, such that two background slots are unlikely to be sampled or 2) any reason why if two background slots are sampled, this does not lead to invalid scenes?
>
> **A2.** We appreciate the comment. In the random mixing strategy, each slot is randomly and independently initialized in each image, and combined into a composite representation by randomly sampling slots from two images. Due to this randomness, there is a chance that mixed slots (*i.e.*, composite representation) lead to ***invalid*** combinations, such as the ones composed only with the objects without background. To avoid this, we share the slot initialization between two images, and mix the slots ***exclusively*** according to their slot indices (*e.g.*, we can sample i-th slot either from the first or second images, but NOT from both). Formally, let $I_1$ and $I_2$ be a random partition of slot indices  *i.e.*, $I_1\\cup I_2=\\{1,...,N\\}, I_1\\cap I_2=\\emptyset$. Then we construct the composite slot by $\\mathbf{S}^c=\\mathbf{S}^1_{I_1}\\cup \\mathbf{S}^2_{I_2}$, where $\\mathbf{S}^1$ and $\\mathbf{S}^2$ are slots extracted from the first and the second images, respectively. Since the slots are initialized identically between two images and the slots extracted from the identical initialization never appear twice in the composite representation, the model can learn to associate certain slot initialization to a specific part of a scene, such that the composite representation is ensured to be always valid. Indeed, we observed that the model learns to associate the background to a certain slot, as shown in Figure 6 in the appendix. We appreciate the comment and will revise the paper to clarify this point.
>
> ---
> >**Q3.**  In ablation study (Table 2), why and how Regularization+Shared Slot Initializations is comparable to using only compositional prior?
>
> **A3.** We would like to first note that both the regularization and compositional loss (generative prior) contribute to enhancing the compositionality of the representation since they are computed from the composition of two images and provide learning signals orthogonal to the auto-encoding path. We observe that the two losses also promote complementary behaviors to the model; The composition loss provides a direct learning signal that the composite image should be realistic, promoting the compositionality of the slot representation. However, we observed that it sometimes captures loose object boundaries especially when the background is monotonic, since they are highly correlated hence compositional. On the other hand, the regularization loss encourages the part of the image captured by a slot to be consistent before and after composition, without considering the realism of the composite image. Combining these two, our method learns to capture compositional components of a scene to a slot while maintaining the consistent association between the slot and a scene, leading to the highest performance.

---

### Official Review · Reviewer_Cp4U · 2023-11-08

**Soundness:** 3 good
**Presentation:** 3 good
**Contribution:** 3 good
**Rating:** 6
**Confidence:** 4

**Summary:**

The paper proposes a new training objective that encourages learning object-centric representation by considering mixtures over objects across images, and empirically explores it in conjunction with a slot attention architecture over CLEVR-like data.

**Update:** Dear authors, thanks for your response, it basically addressed all of my concerns, and I think in particular the additional results on BDD100k strengthen the paper. I would recommend including at least some of them, e.g. some of the qualitative results (a smaller version with less examples of figure 11), in the main paper. Comparison to slot attention is also useful and shows large performance improvement. The comment in the response regarding slot initialization makes it clearer for me now, and I think adding more of it into the paper would be good to make sure it's clear too! Overall, as the response addressed my concerns and presented additional important results, I'm happy to raise my score.

**Strengths:**

- **Evaluation**: Multiple types of experiments are conducted, including quantitative/qualitative unsupervised segmentation, ablation studies, and analysis of robustness. The approach is shown to be robust to factors such as number of slots, as well as settings of the encoder and decoder, aspects which slot attention methods are usually sensitive to.
- **Research Context**: The authors do a good job in providing the relevant research context as well as model’s preliminaries, pointing to the limitations of some of the prior works.
- **Presentation**: The writing quality is good and the paper is clear, well-organized and easy to follow. The first parts of the model section are particularly clear, and both the diagrams and visualizations help understanding the approach and its behavior. The supplementary is also good, providing implementation details and additional quantitative and qualitative results.

**Weaknesses:**

- **Synthetic Data & Scalability**: The experiments are performed over synthetic data only. I recommend exploring scalability to real-world data too. This is especially important since it is unclear to me whether such an approach would scale well to more diverse real-world data, where there are correlations between object occurrence as well as appearance. I don’t know for sure but it might be the case that for realistic data the loss could damage the model’s learning compared to standard auto-encoding, by encouraging it to unlearn important object correlations. For instance, even for the images at figure 1, the lighting of the mixed objects does not fit the context, potentially making the model less aware of the impact of lighting conditions on the object’s appearance. For this reason I believe it is really critical to study this objective in the context of realistic data too.
- **Compositional Synthesis**: The results about compositional synthesis aren't the most compelling. I would like to see object mixing in cases that involve more interesting interactions among the objects (like reflection, more substantial occlusions, and matching of the appearance between the object and its environment). It does perform better than the baselines so that’s still an improvment.
- **Novelty & Related Works**: The specific proposed idea is novel, but object mixing as a form of regularization isn’t. For instance,  it would be good to cite and discuss the “Object Discovery with a Copy-Pasting GAN” paper as a related prior work that also uses mixes over image to encourage object discovery.

**Questions:**

- **Baselines**: It would be good to also compare to a vanilla slot attention as a baseline.
- **Slot Initialization* I didn’t fully understand the intuition why a shared slot initialization should help avoiding bad mixes of objects. It seems to me that learning good combinations of objects could be achieved instead by not sharing the initialization scheme between them and letting them interact across the two images to coordinate valid object compositions. If possible please explain the initialization point further.
- **Diffusion**: It would be helpful to make it clearer how the denoising diffusion and auto-encoding fit together in the proposed framework.
- **Super-CLEVR**: The ARI performance drop on super-clevr is particularly large, even though multiShapenet and PTR also include objects with subparts, and the latter even includes more complicated backgrounds. Do you have an intuition of why is that the case?
- “it decompose” -> “it decomposes”
- “employed auto-encoding framework” -> “employ an auto-encoding framework”
- “as architectural/algorithmic bias” -> “as an architectural/algorithmic bias”
- “atttention” -> “attention”

Overall, while the paper is well-made, I'm in between 5 and 6 due to mentioned weaknesses above. But I'll be glad to raise my score if results will be presented over realistic data.

---

> ### Author Response · Authors · 2023-11-22
> **Official Response to Reviewer Cp4U (1/3)**
>
> We thank the reviewer for the positive comments. Below we respond to the individual questions.
>
> > **Q1.** "The experiments are performed over synthetic data only. I recommend exploring scalability to real-world data too.", "Please provide compositional generation results including interesting interactions among the objects".
>
> **A1.** We appreciate the valuable comments. First, we would like to kindly note that learning object-centric representation in complex real-world images is widely perceived as challenging, and existing works rely on frozen pre-trained encoders and decoders to alleviate the problem. Since freezing the image encoder with a shallow slot attention module limits the opportunity to investigate the true impact of the compositional objective, we follow the standard settings that employ the synthetic data for evaluation.
>
> Nonetheless, we agree with the reviewer that the synthetic scenes are insufficient to demonstrate the scalability of our compositional objective to complex scenes. Following the reviewer’s suggestion, we conducted preliminary experiments on the BDD100K dataset, which contains real-world driving scenes with many objects (cars, signs, buildings, trees, pedestrians, etc.) as well as complex interactions among objects and with the environment (occlusion, shadow, lighting, reflection, etc.). Following the prior work [1], we employed the pre-trained DINOv2 [2] as an image encoder and Stable Diffusion [3] as the generative prior.  We then apply warm-up training for the one-shot decoder for 140K steps with an auto-encoding objective, and apply our compositional objective for the next 100K steps.
>
> The results are summarized in Section B.7 in the Appendix (Figures 10 and 11). Based on visualization of the slot attention map (Figure 10), we observe that our method captures composable object instances not only from the foreground such as cars, buses, and trucks, but also from static environments such as road signs, trees, buildings, sidewalks, etc. In contrast, learning with only diffusion loss often separates an object into multiple slots, or contains multiple objects into a single slot. When composing two images by mixing their slots (Figure 11), we observe that our method successfully generates realistic scenes, modeling complex correlations among objects and environments. It appropriately adapts the appearance of newly added/removed objects, their shadow, reflections in the front glass and hood, and sometimes even global illumination change caused by removing the sun. Our method also generates more accurate compositions compared to the baseline, which often fails to adhere to the addition or removal of the objects.
>
> To summarize, our preliminary experiments showed promising results that the proposed compositional objective can scale to complex scenes and model complex correlations among objects. We believe that incorporating the compositional objective in self-supervised learning can greatly enhance object-centric learning in complex scenes through end-to-end training with the deep encoder, which we plan to investigate in future work.
>
> [1] Jiang et al., Object-centric slot diffusion, NeurIPS 2023.
>
> [2] Oquab et al., DINOv2: Learning robust visual features without supervision, arXiv preprint 2023.
>
> [3] Rombach et al., High resolution image synthesis with latent diffusion models, CVPR 2022.
>
> ---
> > **Q2.** It would be good to cite and discuss the “Object Discovery with a Copy-Pasting GAN” paper as a related prior work that also uses mixes over image to encourage object discovery.
>
> **A2.** We appreciate the valuable comment on the related work. We agree with the reviewers that Copy-Pasting GAN (CP-GAN) shares similar motivation to ours in a spirit of discovering object masks by maximizing the realism of a copy-and-pasted image. Here we outline two major differences as follows. Firstly, our work focuses on learning compositional representation, which can be further employed in downstream tasks or compositional generation tasks, while the CP-GAN focuses on discovering object masks. Secondly, CP-GAN simulates the composition of objects by a simple copy-and-paste on RGB space, while our method combines two scenes using the object (slot) representation and employ powerful diffusion decoder to render the composite scene. We will include these discussions in the paper.

---

> ### Author Response · Authors · 2023-11-22
> **Official Response to Reviewer Cp4U (2/3)**
>
> > **Q3.** It would be good to also compare to a vanilla slot attention as a baseline.
>
> **A3.** We appreciate the comment.  As suggested by the reviewer, we additionally conducted experiments with vanilla slot attention. The results are summarized in the tables below. Overall, the performance of the vanilla slot attention is considerably lower than the other baselines and ours except in the PTR dataset where it achieves slightly better mIoU and mBO than ours. We observe that it is because the slot attention tends to capture tight object boundaries while missing some object parts (*e.g.*, legs of tables), which is favored in mIoU and mBO but results in much lower FG-ARI scores. In all other datasets (CLEVRTex, MultiShapeNet, and Super-CLEVR), however, the vanilla slot attention fails to capture meaningful object masks, often suffering from position bias or leveraging simple color cues due to weak pixel mixture decoder as commonly reported in the literature. We appreciate the comment and will add the results to the main paper.
>
> |        CLEVRTex      | FG-ARI (↑) | mIoU (↑) | mBO (↑) |
> |------------------------|:----------:|:--------:|:-------:|
> | Vanilla Slot Attention |    34.75   |   13.63  |  14.69  |
> |         SLATE+         |    71.29   |   52.04  |  52.17  |
> |           LSD          |    76.44   |   72.32  |  72.44  |
> |          Ours          |    **93.06**   |  **74.82**  |  **75.36**  |
>
>
> |   MultiShapeNet   | FG-ARI (↑) |  mIoU (↑) |  mBO (↑) |
> |------------------------|:----------:|:---------:|:--------:|
> | Vanilla Slot Attention |    69.26   |   16.49   |   16.6   |
> |         SLATE+         |    70.44   |   15.55   |   15.64  |
> |           LSD          |    67.72   |   15.39   |   15.46  |
> |          Ours          |  **89.8**  | **59.21** | **59.4** |
>
>
> |           PTR           | FG-ARI (↑) |  mIoU (↑) |  mBO (↑)  |
> |------------------------|:----------:|:---------:|:---------:|
> | Vanilla Slot Attention |    56.8    | **50.64** | **51.02** |
> |         SLATE+         |  **91.25** |    14.1   |   14.22   |
> |           LSD          |    61.1    |   10.18   |   10.33   |
> |          Ours          |    90.65   |   40.89   |   41.45   |
>
>
> |    Super-CLEVR    | FG-ARI (↑) |  mIoU (↑) |  mBO (↑)  |
> |------------------------|:----------:|:---------:|:---------:|
> | Vanilla Slot Attention |    26.92   |   13.63   |   14.69   |
> |         SLATE+         |    43.73   |   29.12   |   29.49   |
> |           LSD          |    54.79   |   14.12   |   14.43   |
> |          Ours          |  **63.08** | **47.17** | **48.03** |
> ---
> > **Q4.** Please elaborate more on why sharing slot initialization helps avoiding bad mixing of objects?
>
> **A4.** We appreciate the comment. In the random mixing strategy, each slot is randomly and independently initialized in each image, and combined into a composite representation by randomly sampling slots from two images. Due to this randomness, there is a chance that mixed slots (*i.e.*, composite representation) lead to ***invalid*** combinations, such as the ones composed only with the objects without background. To avoid this, we share the slot initialization between two images, and mix the slots ***exclusively*** according to their slot indices (*e.g.*, we can sample i-th slot either from the first or second images, but NOT from both). Formally, let $I_1$ and $I_2$ be a random partition of slot indices *i.e.*, $I_1\cup I_2=\\{ 1,...,N\\}, I_1\cap I_2=\\emptyset$. Then we construct the composite slot by $\\mathbf{S}^c=\\mathbf{S}^1_{I_1}\\cup \\mathbf{S}^2_{I_2}$, where $\\mathbf{S}^1$ and $\\mathbf{S}^2$ are slots extracted from the first and the second images, respectively. Since the slots are initialized identically between two images and the slots extracted from the identical initialization never appear twice in the composite representation, the model can learn to associate certain slot initialization to a specific part of a scene, such that the composite representation is ensured to be always valid. Indeed, we observed that the model learns to associate the background to a certain slot, as shown in Figure 6 in the appendix. We appreciate the comment and will revise the paper to clarify this point.

---

> ### Author Response · Authors · 2023-11-22
> **Official Response to Reviewer Cp4U (3/3)**
>
> > **Q5.** Please clarify how the denoising diffusion and auto-encoding fit together in the proposed framework.
>
> **A5.** Our diffusion decoder is trained with auto-encoding loss (Eq.(4) in the paper) while also used to compute the composition loss (Eq.(6)). In the auto-encoding path, the diffusion decoder learns the marginal distribution of data, while in the composition path, it is used to evaluate the likelihood of the composite image. We apply the stop gradient to the diffusion decoder in the composition path to ensure that it learns with only the true data distribution through the auto-encoding path. We will clarify the paper to make it more clear.
>
> ---
> > **Q6.** Why does the ARI performance on Super-CLEVR drop a lot compared to other datasets? Please provide an intuition on this.
>
> **A6.** The ground-truth object masks of the Super-CLEVR dataset are much more fine-grained than the other datasets. Since the resolution of the attention map in our method and baselines is much coarser than the mask, we observe that it generally degrades the FG-ARI score for all baselines (*e.g.*, a small error in a low-resolution attention map can lead to a bigger penalty), especially when the objects are occluded by the others (*e.g.*, occluded by the spoke of a bike).

---

> > ### Comment · Reviewer_Cp4U · 2023-12-04
> > **Thanks for the response**
> >
> > Dear authors, thanks for your response, it basically addressed all of my concerns, and I think in particular the additional results on BDD100k strengthen the paper. I would recommend including at least some of them, e.g. some of the qualitative results (a smaller version with less examples of figure 11), in the main paper. Comparison to slot attention is also useful and shows large performance improvement. The comment in the response regarding slot initialization makes it clearer for me now, and I think adding more of it into the paper would be good to make sure it's clear too! Overall, as the response addressed my concerns and presented additional important results, I'm happy to raise my score.

---

### Meta-Review · Area_Chair_iEXX · 2023-12-05

**Metareview:**

This paper proposes a new training objective that encourages compositionality in slot-based models for object-centric representation learning. It works by mixing subsets of slots from two separate scenes, and training the model to produce a plausible new scene from under the data distribution. The plausibility of the new scene is assessed via a generative prior derived from the same diffusion model that is used in the usual reconstruction path. Improvements for unsupervised object discovery are reported on several standard (albeit mostly synthetic) benchmarks. It is also shown how the proposed regularizer helps with model robustness to hyper-parameters such as varying the number of slots, or changing the network capacity.

This paper initially received mixed scores. Reviewers pointed out how the experiments are thorough and convincing, the proposed approach is clever and makes sense, and the overall approach is well-motivated. However, there were some concerns regarding the simplicity of the benchmarks (only considering synthetic data), which were addressed in the rebuttal with new results on BDD. Additional comparisons that were requested, such as to vanilla Slot Attention and DINOSAUR, were also provided. In response, reviewers have increased their scores and are now in broad agreement that this work is (marginally) above the acceptance threshold.

Overall I agree with the reviewer assessment of this work. The proposed idea is simple and well executed. The use of a diffusion decoder both for reconstruction and as a generative prior is interesting and certainly new among methods for object-centric learning. On the flipside, mixing content from different images has previously been explored (eg. Copy-Paste GAN), and though the results are solid given the benchmarks that have typically been explored, they are not spectacular either. The new BDD results are promising, though quantitative results are missing.

**Justification For Why Not Higher Score:**

Despite the execution of this work, the significance of the presented results are limited to a small community, and even within that community it is hard to argue that this work makes significant progress toward more real-world datasets (which is one of the main issues today). The BDD results are a step in this direction, yet they are quantitative only as far as I can tell.

**Justification For Why Not Lower Score:**

The reviewers all agree that the current contribution is sufficient to warrant acceptance. The paper is well executed, the method is clear and intuitive, and there is some novelty regarding the use of a diffusion model for both scoring of novel scene compositions and for reconstruction.

---

### Decision · Program_Chairs · 2024-01-16

Accept (poster)